# Firing rate adaptation affords place cell theta sweeps, phase precession, and procession

Tianhao Chu[1†], Zilong Ji[1,2†], Junfeng Zuo[1], Yuanyuan Mi[3], Wen-hao Zhang[4], Tiejun Huang[5], Daniel Bush[6], Neil Burgess[2], Si Wu[1]*

[1]School of Psychological and Cognitive Sciences, IDG/McGovern Institute for Brain Research, Center of Quantitative Biology, Peking-Tsinghua Center for Life Sciences, Academy for Advanced Interdisciplinary Studies, Peking University, Beijing, China; [2]Institute of Cognitive Neuroscience, University College London, London, United Kingdom; [3]Department of Psychology, Tsinghua University, Beijing, China; [4]Lyda Hill Department of Bioinformatics, O'Donnell Brain Institute, The University of Texas Southwestern Medical Center, Dallas, United States; [5]School of Computer Science, Peking University, Beijing, China; [6]Department of Neuroscience, Physiology and Pharmacology, University College London, London, United Kingdom

*For correspondence:
siwu@pku.edu.cn

†These authors contributed equally to this work

Competing interest: The authors declare that no competing interests exist.

**Abstract** Hippocampal place cells in freely moving rodents display both theta phase precession and procession, which is thought to play important roles in cognition, but the neural mechanism for producing theta phase shift remains largely unknown. Here, we show that firing rate adaptation within a continuous attractor neural network causes the neural activity bump to oscillate around the external input, resembling theta sweeps of decoded position during locomotion. These forward and backward sweeps naturally account for theta phase precession and procession of individual neurons, respectively. By tuning the adaptation strength, our model explains the difference between 'bimodal cells' showing interleaved phase precession and procession, and 'unimodal cells' in which phase precession predominates. Our model also explains the constant cycling of theta sweeps along different arms in a T-maze environment, the speed modulation of place cells' firing frequency, and the continued phase shift after transient silencing of the hippocampus. We hope that this study will aid an understanding of the neural mechanism supporting theta phase coding in the brain.

## eLife assessment

This study provides **valuable** new insights on how a prevailing model of hippocampal sequence formation can account for recent data, including forward and backward sweeps, as well as constant cycling of sweeps across different arms of a T-maze. The **convincing** evidence presented in support of this work relies on classical analytical and computational techniques about continuous attractor networks.

## Introduction

One of the strongest candidates for temporal coding of a cognitive variable by neural firing is the 'theta phase precession' shown by hippocampal place cells. As an animal runs through the firing field of a place cell, the cell fires at progressively earlier phases in successive cycles of the ongoing local field potential (LFP) theta oscillation, so that firing phase correlates with distance traveled (*O'Keefe and Recce, 1993*; *Skaggs et al., 1996*; see also *Schmidt et al., 2009*; *Figure 1a and b*). At the

**Figure 1.** Theta sequence and theta phase shift of place cell firing. (**a**) An illustration of an animal running on a linear track. A group of place cells each represented by a different color are aligned according to their firing fields on the linear track. (**b**) An illustration of the forward theta sequences of the neuron population (upper panel), and the theta phase precession of the fourth place cell (represented by the green color, lower panel). (**c**) An illustration of both forward and reverse theta sequences (upper panel), and the corresponding theta phase precession and procession of the fourth place cell (lower panel). The sinusoidal trace illustrates the theta rhythm of local field potential (LFP), with individual theta cycles separated by vertical dashed lines.

population level, phase precession of individual cells gives rise to forward theta sequences once starting phases are aligned across the population (*Feng et al., 2015*), where neurons representing successive locations along the trajectory of the animal display predictable firing sequences within individual theta cycles (*Johnson and Redish, 2007*). These prospective sequential experiences (looking into the future) are potentially useful for a range of cognitive faculties, e.g., planning, imagination, and decision-making (*O'Keefe and Recce, 1993*; *Skaggs et al., 1996*; *Hassabis et al., 2007*; *Wikenheiser and Redish, 2015*; *Kay et al., 2020*).

Besides prospective representation, flexible behaviors also require retrospective representation of sequential experiences (looking into the past). For instance, in goal-directed behaviors, it is important to relate the reward information that might only occur at the end of a sequence of events to preceding events in the sequence (*Foster et al., 2000*; *Foster and Wilson, 2006*; *Diba and Buzsáki, 2007*). A recent experimental study (*Wang et al., 2020*) described retrospective sequences during online behaviors (also indicated by *Skaggs et al., 1996*; *Yamaguchi et al., 2002*), namely, reverse theta sequences, interleaved with forward theta sequences in individual theta cycles (*Figure 1c*). Such retrospective sequences, together with the prospective sequences, may cooperate to establish higher-order associations in episodic memory (*Diba and Buzsáki, 2007*; *Jaramillo and Kempter, 2017*; *Pfeiffer, 2020*).

While a large number of computational models of phase precession and the associated forward theta sequences have been proposed, e.g., the single-cell oscillatory models (*O'Keefe and Recce, 1993*; *Kamondi et al., 1998*; *Harris et al., 2002*; *Lengyel et al., 2003*; *Losonczy et al., 2010*) and recurrent activity spreading models (*Tsodyks et al., 1996*; *Romani and Tsodyks, 2015*), the underlying neural mechanism for interleaved forward- and reverse-ordered sequences remains largely unclear. Do reverse theta sequences share the same underlying neural mechanism as forward sequences, or do they reflect different mechanisms? If they do, what kind of neural architecture can support the emergence of both kinds of theta phase shift? Furthermore, since forward theta sequences are commonly seen, but reverse theta sequences are only seen in some circumstances (*Wang et al., 2020*), are they

commensurate with forward theta sequences? If not, to what degree are forward theta sequences more significant than the reverse ones?

To address these questions, we built a continuous attractor neural network (CANN) of the hippocampal place cell population (*Amari, 1977*; *Tsodyks and Sejnowski, 1995*; *Samsonovich and McNaughton, 1997*; *Tsodyks, 1999*). The CANN conveys a map of the environment in its recurrent connections that affords a single bump of activity on a topographically organized sheet of cells which can move smoothly so as to represent the location of the animal as it moves in the environment. Each neuron exhibits firing rate adaptation which destabilizes the bump attractor state. When the adaptation is strong enough, the network bump can travel spontaneously in the attractor space, which we term as the intrinsic mobility. Intriguingly, we show that, under competition between the intrinsic mobility and the extrinsic mobility caused by location-dependent sensory inputs, the network displays an oscillatory tracking state, in which the network bump sweeps back and forth around the external sensory input. This phenomenon naturally explains the theta sweeps found in the hippocampus (*Skaggs et al., 1996*; *Burgess et al., 1994*; *Foster and Wilson, 2007*), where the decoded position sweeps around the animal's physical position at theta frequency. More specifically, phase precession occurs when the bump propagates forward while phase procession occurs when the network bump propagates backward. Moreover, we find that neurons can exhibit either only predominant phase precession (unimodal cells) when adaptation is relatively strong, or interleaved phase precession and procession (bimodal cells) when adaptation is relatively weak.

In addition to theta phase shift, our model also successfully explains the constant cycling of theta sweeps along different upcoming arms in a T-maze environment (*Kay et al., 2020*), and other phenomena related to phase precession of place cells (*Geisler et al., 2007*; *Zugaro et al., 2005*). We hope that this study facilitates our understanding of the neural mechanism underlying the rich dynamics of hippocampal neurons and lays the foundation for unveiling their computational functions.

## Results

### A network model of hippocampal place cells

To study the phase shift of hippocampal place cells, we focus on a one-dimensional (1D) CANN (mimicking the animal moving on a linear track, see *Figure 2a*), but generalization to the 2D case (mimicking the animal moving in a 2D arena) is straightforward (see Discussion for more details). Neurons in the 1D CANN can be viewed as place cells rearranged according to the locations of their firing fields on the linear track (measured during free exploration). The dynamics of the 1D CANN is written as:

$$\tau \frac{dU(x,t)}{dt} = -U(x,t) + \rho \int_{-\infty}^{\infty} J(x,x')r(x',x)dx' - V(x,t) + I^{ext}(x,t), \tag{1}$$

$$r(x,t) = \frac{gU(x,t)^2}{1 + k\rho \int_{-\infty}^{\infty} U^2(x',t)\,dx'}, \tag{2}$$

Here, $U(x,t)$ represents the presynaptic input to the neuron located at position $x$ on the linear track, and $r(x,t)$ represents the corresponding firing rate constrained by global inhibition (*Hao et al., 2009*). $\tau$ is the time constant, $\rho$ the neuron density, $k$ the global inhibition strength, and $g$ is the gain factor. The dynamics of $U(x,t)$ is determined by the leaky term $-U(x,t)$, the recurrent input from other neurons, the firing rate adaptation $-V(x,t)$, and the external input $I^{ext}(x,t)$. The recurrent connection strength $J(x,x')$ between two neurons decays with their distance. For simplicity, we set $J(x,x')$ to be the Gaussian form, i.e., $J(x,x') = J_0/\sqrt{2\pi a}exp\left[-(x-x')^2/(2a^2)\right]$, with $J_0$ controlling the connection strength and $a$ the range of neuronal interaction. Such connectivity gives rise to a synaptic weight matrix with the property of translation invariance. Together with the global inhibition, the translation invariant weight matrix ensures that the network can hold a continuous family of stationary states (attractors) when no external input and adaptation exist (*Tsodyks and Sejnowski, 1995*; *Samsonovich and McNaughton, 1997*; *McNaughton et al., 2006*; *Wu et al., 2008*), where each attractor is a localized firing bump representing a single spatial location (*Figure 2b*). These bump states are expressed as (see 'Stability analysis of the bump state' for the parameter settings and 'Deriving the network state when the external input does not exist ($I^{ext}$ = 0)' for the detailed mathematical derivation):

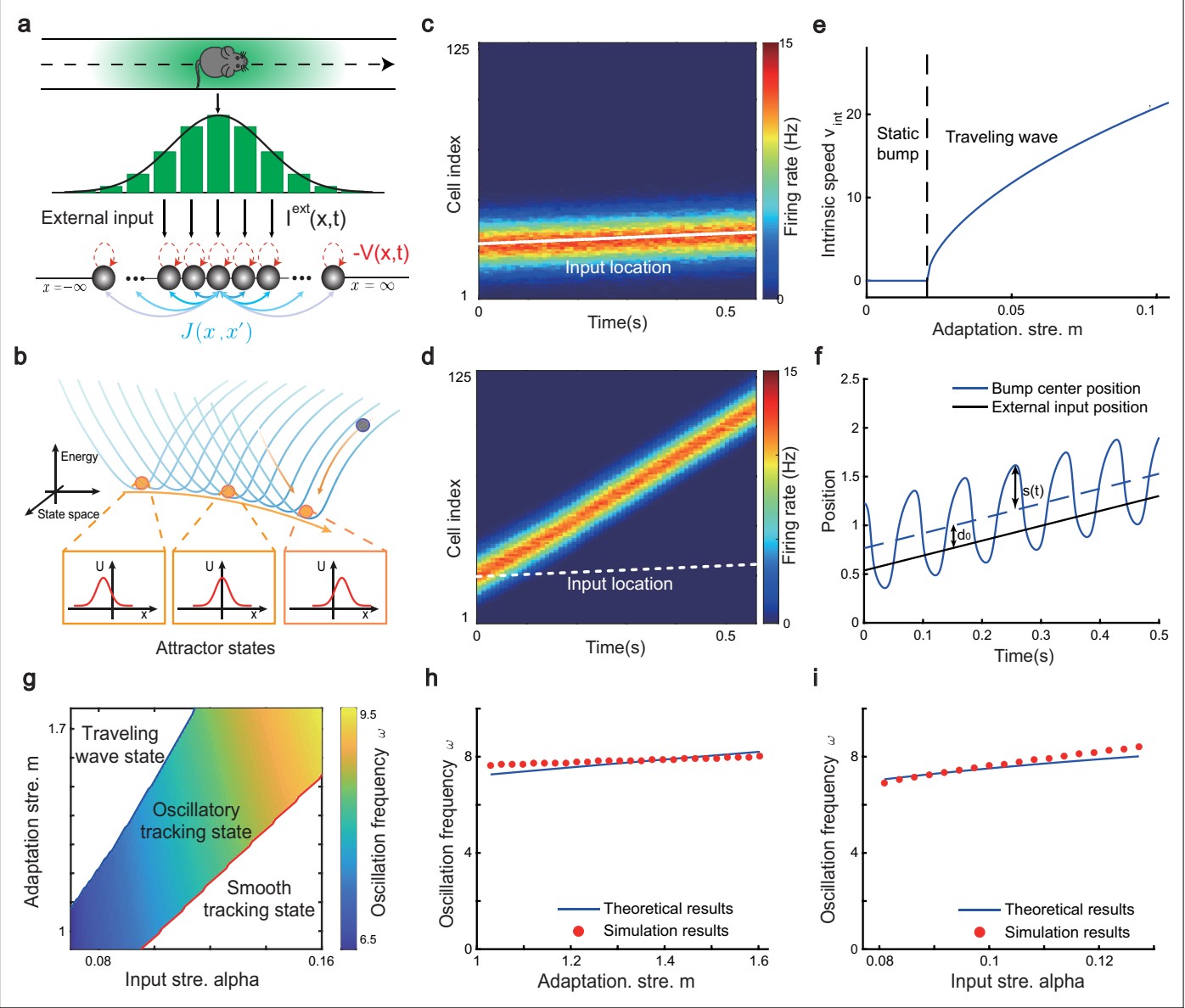

**Figure 2.** The network architecture and tracking dynamics. (**a**) A one-dimensional (1D) continuous attractor neural network (CANN) formed by place cells. Neurons are aligned according to the locations of their firing fields on the linear track. The recurrent connection strength $J(x, x')$ (blue arrows) between two neurons decays with their distance on the linear track. Each neuron receives an adaptation current $-V(x, t)$ (red dashed arrows). The external input $I^{ext}(x, t)$, represented by a Gaussian-shaped bump, conveys location-dependent sensory inputs to the network. (**b**) An illustration of the state space of the CANN. The CANN holds a family of bump attractors which form a continuous valley in the energy space. (**c**) The smooth tracking state. The network bump (hot colors) smoothly tracks the external moving input (the white line). The red (blue) color represents high (low) firing rate. (**d**) The traveling wave state when the CANN has strong firing rate adaptation. The network bump moves spontaneously with a speed much faster than the external moving input. (**e**) The intrinsic speed of the traveling wave versus the adaptation strength. (**f**) The oscillatory tracking state. The bump position sweeps around the external input (black line) with an offset $d_0$. (**g**) The phase diagram of the tracking dynamics with respect to the adaptation strength $m$ and the external input strength $\alpha$. The colored area shows the parameter regime for the oscillatory tracking state. Yellow (blue) color represents fast (slow) oscillation frequency. (**h and i**) Simulated (red points) and theoretical (blue line) oscillation frequency as a function of the adaptation strength (**h**) or the external input strength (**i**).

$$\bar{r}(x,t) = A_r(t) \, exp \left\{ -\frac{[x - z(t)]^2}{2a^2} \right\}, \tag{3}$$

where $A_r(t)$ denotes the bump height and $z(t)$ the bump center, i.e., the spatial location represented by the network. For convenience, we set the external input to be of the Gaussian form, which is written as: $I^{ext}(x,t) = \alpha \, exp \left[ -(x - v_{ext}t)^2 / \left( 4a^2 \right) \right]$, with $v_{ext}$ representing the moving speed and $\alpha$ controlling the external input strength. Such external moving input represents location-dependent sensory inputs (i.e. corresponding to the animal's physical location) which might be conveyed via the entorhinal-hippocampal or subcortical pathways (*van Strien et al., 2009*). The term $-V(x,t)$ represents the firing rate adaptation (*Alonso and Klink, 1993*; *Fuhrmann et al., 2002*; *Benda and Herz, 2003*; *Treves, 2004*), whose dynamics is written as:

$$\tau_v \frac{dV(x,t)}{dt} = -V(x,t) + mU(x,t), \tag{4}$$

where $m$ controls the adaptation strength, and $\tau_v$ is the time constant. The condition $\tau_v \gg \tau$ holds, implying that the firing rate adaptation is a much slower process compared to neuronal firing. In effect, the firing rate adaptation increases with the neuronal activity and contributes to destabilizing the active bump state, which induce rich dynamics of the network (see below).

## Oscillatory tracking of the network

Overall, the bump motion in the network is determined by two competing factors, i.e., the external input and the adaptation. The interplay between these two factors leads to the network exhibiting oscillatory tracking in an appropriate parameter regime. To elucidate the underlying mechanism clearly, we explore the effects of the external input and the adaptation on bump motion separately. First, when firing rate adaptation does not exist in the network ($m = 0$), the bump tracks the external moving input smoothly (see *Figure 2c*). We refer to this as the '**smooth tracking state**', where the internal location represented in the hippocampus (the bump position) is continuously tracking the animal's physical location (the external input location). This smooth tracking property of CANNs has been widely used to model spatial navigation in the hippocampus (*Tsodyks and Sejnowski, 1995*; *Samsonovich and McNaughton, 1997*; *McNaughton et al., 2006*; *Battaglia and Treves, 1998*). Second, when the external drive does not exist in the network ($\alpha = 0$) and the adaptation strength $m$ exceeds a threshold ($m > \tau/\tau_v$), the bump moves spontaneously with a speed calculated as $v_{int} = (2a/\tau_v) \sqrt{m\tau_v/\tau - \sqrt{m\tau_v/\tau}}$ (see *Figure 2d and e* and 'Analysis of the intrinsic mobility of the bump state' for more details). We refer to this as the '**traveling wave state**', where the internal representation of location in the hippocampus is sequentially reactivated without external drive, resembling replay-like dynamics during a quiescent state (see Discussion for more details). This intrinsic mobility of the bump dynamics can be intuitively understood as follows. Neurons around the bump center have the highest firing rates and hence receive the strongest adaptation. Such strong adaptation destabilizes the bump stability at the current location, and hence pushes the bump away. After moving to a new location, the bump will be continuously pushed away by the firing rate adaptation at the new location. As a result, the bump keeps moving on the linear track. Similar mechanisms have been applied to explain mental exploration (*Hopfield, 2010*), preplay during sharp wave-ripple events in the hippocampus (*Azizi et al., 2013*), and the free memory recall phenomenon in the brain (*Dong et al., 2021*).

When both the external input and adaptation are applied to the CANN, the interplay between the extrinsic mobility (caused by the external input) and the intrinsic mobility (caused by the adaptation) will induce three different dynamical behaviors of the network (see *Video 1* for demonstration), i.e., (1) when $m$ is small and $\alpha$ is large, the network displays the smooth tracking state; (2) when $m$ is large and $\alpha$ is small, the network displays the traveling wave state; (3) when both $m$

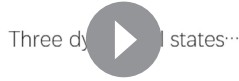

Three dynamical states⋯

**Video 1.** The title of this video is: Three dynamical states of Adaptive Continous Attractor Neural Network.
https://elifesciences.org/articles/87055/figures#video1

and $\alpha$ have moderate values, the network bump displays an interesting state, called the '**oscillatory tracking state**', where the bump tracks the external moving input in an oscillatory fashion (*Figure 2f and g*). Intuitively, the mechanism for oscillatory tracking can be understood as follows. Due to the intrinsic mobility of the network, the bump tends to move at its own intrinsic speed (which is faster than the external moving input, see *Figure 2d*), i.e., the bump tries to escape from the external input. However, due to the strong locking effect of the external input, the bump cannot run too far away from the location input, but instead, is attracted back to the location input. Once the bump returns, it will keep moving in the opposite direction of the external input until it is pulled back by the external input again. Over time, the bump will sweep back and forth around the external moving input, displaying the oscillatory tracking behavior. It is noteworthy that the activity bump does not live within a window circumscribed by the external input bump (bouncing off the interior walls of the input during the oscillatory tracking state), but instead is continuously pulled back and forth by the external input (see *Appendix 1—figure 1*).

Our study shows that during oscillatory tracking, the bump shape is roughly unchanged (see previous sections for the condition of shape variability), and the bump oscillation can be well represented as the bump center sweeping around the external input location. The dynamics of the bump center can be approximated as a propagating sinusoidal wave (*Figure 2f*), i.e.,

$$z\left(t\right) = c_0 sin\left(\omega t\right) + d_0 + v_{ext}t = s\left(t\right) + v_{ext}t, \tag{5}$$

where $z\left(t\right)$ is the bump center at time $t$ (see *Equation 3*). $s\left(t\right)$ denotes the displacement between the bump center and the external input, which oscillates at the frequency $\omega$ with the amplitude $c_0 > 0$ and a constant offset $d_0 > 0$ (see 'Analysis of the oscillatory tracking behavior of the bump state' for the values of these parameters and 'Deriving the oscillatory tracking state of the network when the external input is applied ($I^{ext} \neq 0$)' for the detailed derivation). When the firing rate adaptation is relatively small, the bump oscillation frequency can be analytically solved to be (see also *Appendix 1—figure 2*):

$$\omega = \sqrt{\frac{2\sqrt{\pi}\alpha ak\left(1+m\right)}{\tau\tau_v\left(J_0 + 2\sqrt{\pi}ak\alpha\right)}}. \tag{6}$$

We see that the bump oscillation frequency $\omega$ increases sublinearly with the external input strength $\alpha$ and the adaptation strength $m$ (*Figure 2h and i*). By setting the parameters appropriately, the bump can oscillate in the theta band (6–10 Hz), thus approximating the experimentally observed theta sweeps (see below). Notably, LFP theta is not explicitly modeled in the network. However, since theta sweeps are bounded by individual LFP theta cycles in experiments, they share the same oscillation frequency as LFP theta. For convenience, we will frequently use the term LFP theta below and study firing phase shift in individual oscillation cycles.

## Oscillatory tracking accounts for both theta phase precession and procession of hippocampal place cells

In our model, the bump center and external input represent the decoded and physical positions of the animal, respectively, thus the oscillatory tracking of the bump around the external input naturally gives rise to the forward and backward theta sweeps observed empirically (*Figure 3a and b*; *Wang et al., 2020*). Here, we show that oscillatory tracking of the bump accounts for the theta phase precession and procession of place cell firing.

Without loss of generality, we select the neuron at location $x = 0$ as the probe neuron and examine how its firing phase changes as the external input traverses its firing field (*Figure 3c*). In the absence of explicitly simulated spike times, the firing phase of a neuron in each theta cycle is measured by the moment when the neuron reaches the peak firing rate (see 'Spike generation from the firing rate' for modeling spike times in the CANN). Based on *Equations 3 and 5*, the firing rate of the probe neuron, denoted as $r_0\left(t\right)$, is expressed as:

$$r_0\left(t\right) = A_r\left(t\right) exp\left[-\frac{\left[0 - z\left(t\right)\right]^2}{2a^2}\right] = A_r\left(t\right) exp\left[-\frac{\left(v_{ext}t + c_0 sin\omega t + d_0\right)^2}{2a^2}\right] \equiv A_r\left(t\right) exp\left[-\frac{h\left(t\right)^2}{2a^2}\right], \tag{7}$$

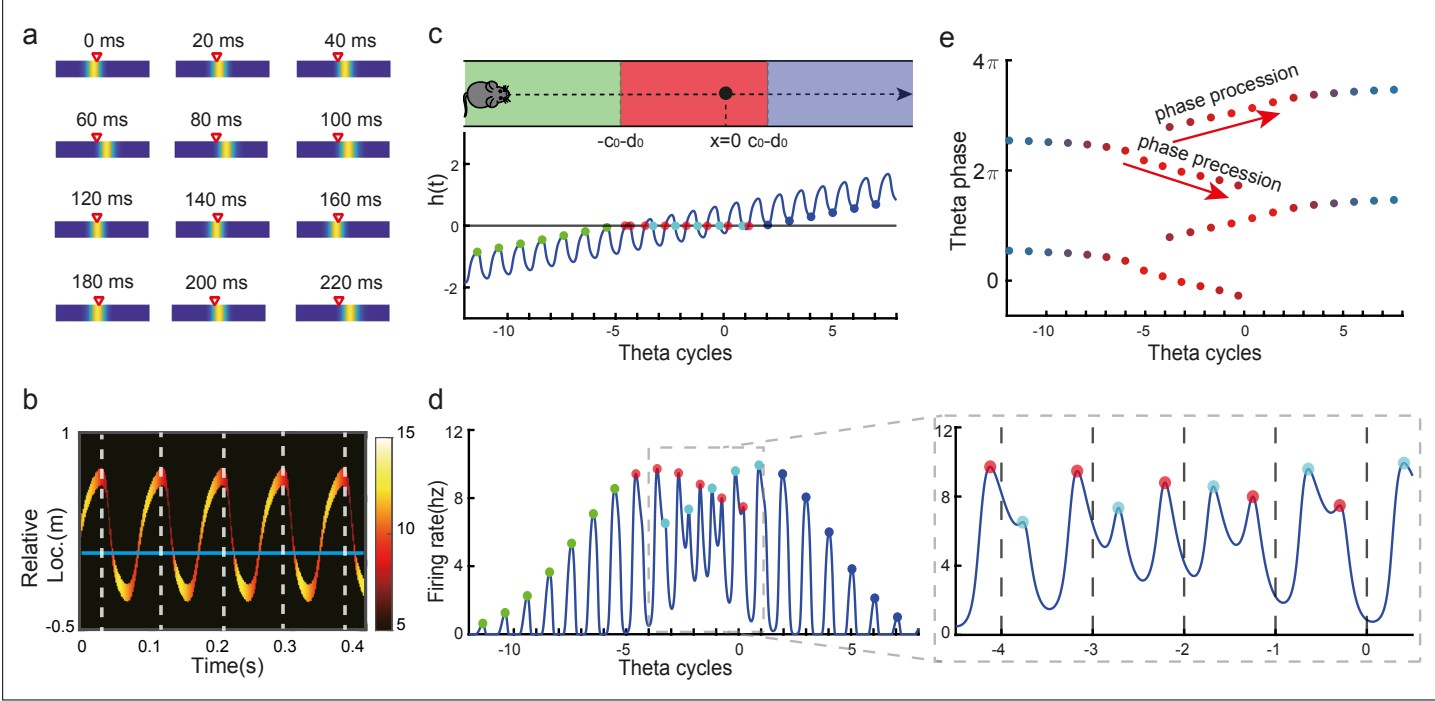

**Figure 3.** Oscillatory tracking accounts for theta sweeps and theta phase shift. (**a**) Snapshots of the bump oscillation along the linear track in one theta cycle (0–140 ms). Red triangles indicate the location of the external moving input. (**b**) Decoded relative positions based on place cell population activities. The relative locations of the bump center (shown by the neural firing rates of 10 most active neurons at each timestamp) with respect to the location of the external input (horizontal line) in five theta cycles. See a comparison with experimental data in *Wang et al., 2020*, Figure 1a lower panel. (**c**) Upper panel: The process of the animal running through the firing field of the probe neuron (large black dot) is divided into three stages: the entry stage (green), the phase shift stage (red), and the departure stage (blue). Lower panel: The displacement between the bump center and the probe neuron as the animal runs through the firing field. The horizontal line represents the location of the probe neuron, which is $x = 0$. (**d**) The firing rates of the probe neuron as the animal runs through the firing field. Colored points indicate firing peaks. The trace of the firing rate in the phase shift stage (the dashed box) is enlarged in the sub-figure on the right-hand side, which exhibits both phase precession (red points) and procession (blue points) in successive theta cycles. (**e**) The firing phase shift of the probe neuron in successive theta cycles. Red points progress to earlier phases from $\pi/2$ to $-\pi/2$ and blues points progress to later phases from $\pi/2$ to $3\pi/2$. The color of the dots represent the peak firing rates, which is also shown in (d).

where $A_r(t)$ is the bump height, and $h(t)$ is an oscillatory moving term denoting the displacement between the bump center and the location of the probe neuron. It is composed of a moving signal $v_{ext}t$ and an oscillatory signal $c_0 sin\omega t + d_0$, with $c_0$ the oscillation amplitude, $\omega$ the frequency, and $d_0$ an oscillation offset constant. It can be seen that the firing rate of the probe neuron is determined by two factors, $A_r(t)$ and $h(t)$. To simplify the analysis below, we assume that the bump height $A_r(t)$ remains unchanged during bump oscillations (for the case of time-varying bump height, see previous sections). Thus, the firing rate only depends on $h(t)$, which is further determined by two time-varying terms, the oscillation term $c_0 sin\omega t$ and the location of the external input $v_{ext}t$. The first term contributes to firing rate oscillations of the probe neuron, and the second term contributes to the envelope of neuronal oscillations exhibiting a waxing-and-waning profile over time, as the external input traverses the firing field (the absolute value $|v_{ext}t|$ first decreases and then increases; see *Figure 3d*, also *Video 2*). Such a waxing-and-waning profile agrees well with the experimental data (*Skaggs et al., 1996*). In each LFP theta cycle, the peak firing rate of the probe neuron is achieved when $|h(t)|$ reaches a local minima (*Figure 3c and d*). We differentiate three stages as the external input

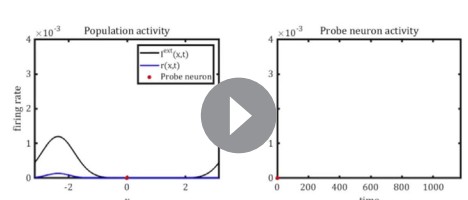

**Video 2.** The title of this video is: Neuronal activities during bi-directional oscillatory tracking state.
https://elifesciences.org/articles/87055/figures#video2

passes through the probe neuron (i.e. the animal travels through the place field of the probe neuron), i.e.,

- The entry stage. As the external input enters the firing field of the probe neuron (moving from left to right), $h(t) < 0$ always holds (**Figure 3c**). In this case, the peak firing rate of the probe neuron in each oscillatory cycle is achieved when $h(t)$ reaches the maximum (i.e. $|h(t)|$ reaches the minimum). This corresponds to $c_0 sin\omega t = c_0$, i.e., $\omega t = \pi/2$ (**Figure 3e**). This means that the firing phase of the probe neuron at the entry stage is constant, which agrees with experimental observations (**O'Keefe and Recce, 1993**; **Skaggs et al., 1996**).
- The phase shift stage. As the external input moves into the center of the firing field, $h(t) = 0$ can be achieved in each oscillatory cycle (**Figure 3c**). Notably, it is achieved twice in each cycle, once as the bump sweeps over the probe neuron in the forward direction and the other as the bump sweeps over the probe neuron in the backward direction. Therefore, there are two firing peaks in each bump oscillation cycle (**Figure 3d**), which are expressed as (by solving $v_{ext}t + c_0 sin\omega t + d_0 = 0$):

$$\phi_f = -arcsin\left[\frac{d_0 + v_{ext}t_f}{c_0}\right], \quad \phi_b = \pi + arcsin\left[\frac{d_0 + v_{ext}t_b}{c_0}\right], \tag{8}$$

where $t_f$ and $t_b$ denote the moments of peak firing in the forward and backward sweeps, respectively, and $\phi_f$ and $\phi_b$ the corresponding firing phases of the probe neuron. As the external input travels from $(-c_0 - d_0)$ to $(c_0 - d_0)$, the firing phase $\phi_f$ in the forward sweep decreases from $\pi/2$ to $-\pi/2$, while the firing phase $\phi_r$ in the backward sweep increases from $\pi/2$ to $3\pi/2$ (**Figure 3e**). These give rise to the phase precession and procession phenomena, respectively, agreeing well with experimental observations (**Skaggs et al., 1996**; **Wang et al., 2020**; **Yamaguchi et al., 2002**).

- The departure stage. As the external input leaves the firing field, $h(t) > 0$ always holds (**Figure 3c**), and the peak firing rate of the probe neuron is achieved when $h(t)$ reaches its minimum in each oscillatory cycle, i.e., $c_0 sin(\omega t) = -c_0$ with $\omega t = \pi/2$ (**Figure 3e**). Therefore, the firing phase of the probe neuron is also constant during the departure stage.

In summary, oscillatory tracking of the CANN well explains the firing phase shift of place cells when the animal traverses their firing fields. Specifically, when the animal enters the place field, the firing phase of the neuron remains constant, i.e., no phase shift occurs, which agrees with experimental observations (**O'Keefe and Recce, 1993**; **Skaggs et al., 1996**). As the animal approaches the center of the place field, the firing phase of the neuron starts to shift in two streams, one to earlier phases during the forward sweeps and the other to later phases during the backward sweeps. Finally, when the animal leaves the place field, the firing phase of the neuron stops shifting and remains constant. Over the whole process, the firing phase of a place cell is shifted by 180 degrees, which agrees with experimental observations (**O'Keefe and Recce, 1993**; **Skaggs et al., 1996**).

## Different adaptation strengths account for bimodal and unimodal cells

The results above show that during oscillatory tracking, a place cell exhibits both significant phase precession and procession, which are associated with two firing peaks in a theta cycle. These neurons have been described as bimodal cells (**Wang et al., 2020**; **Figure 4a**). Conversely, previous experiments have primarily focused on the phase precession of place cell firing, while tending to ignore phase procession, which is a relatively weaker phenomenon (**O'Keefe and Recce, 1993**; **Skaggs et al., 1996**). Place cells with negligible phase procession have been described as unimodal cells (**Figure 4b**).

Here, we show that by adjusting a single parameter in the model, i.e., the adaptation strength $m$, neurons in the CANN can exhibit either interleaved phase precession and procession (bimodal cells) or predominant phase precession (unimodal cells). To understand this, we first recall that the firing rate adaptation is a much slower process compared to neural firing and its timescale is in the same order as the LFP theta (i.e. $\tau_v = 100$ ms while $\tau = 5$ ms). This implies that when the bump sweeps over a neuron, the delayed adaptation it generates will suppress the bump height as it sweeps back to the same location. Furthermore, since the oscillatory tracking always begins with a forward sweep (as the initial sweep is triggered by the external input moving in the same direction), the suppression effects

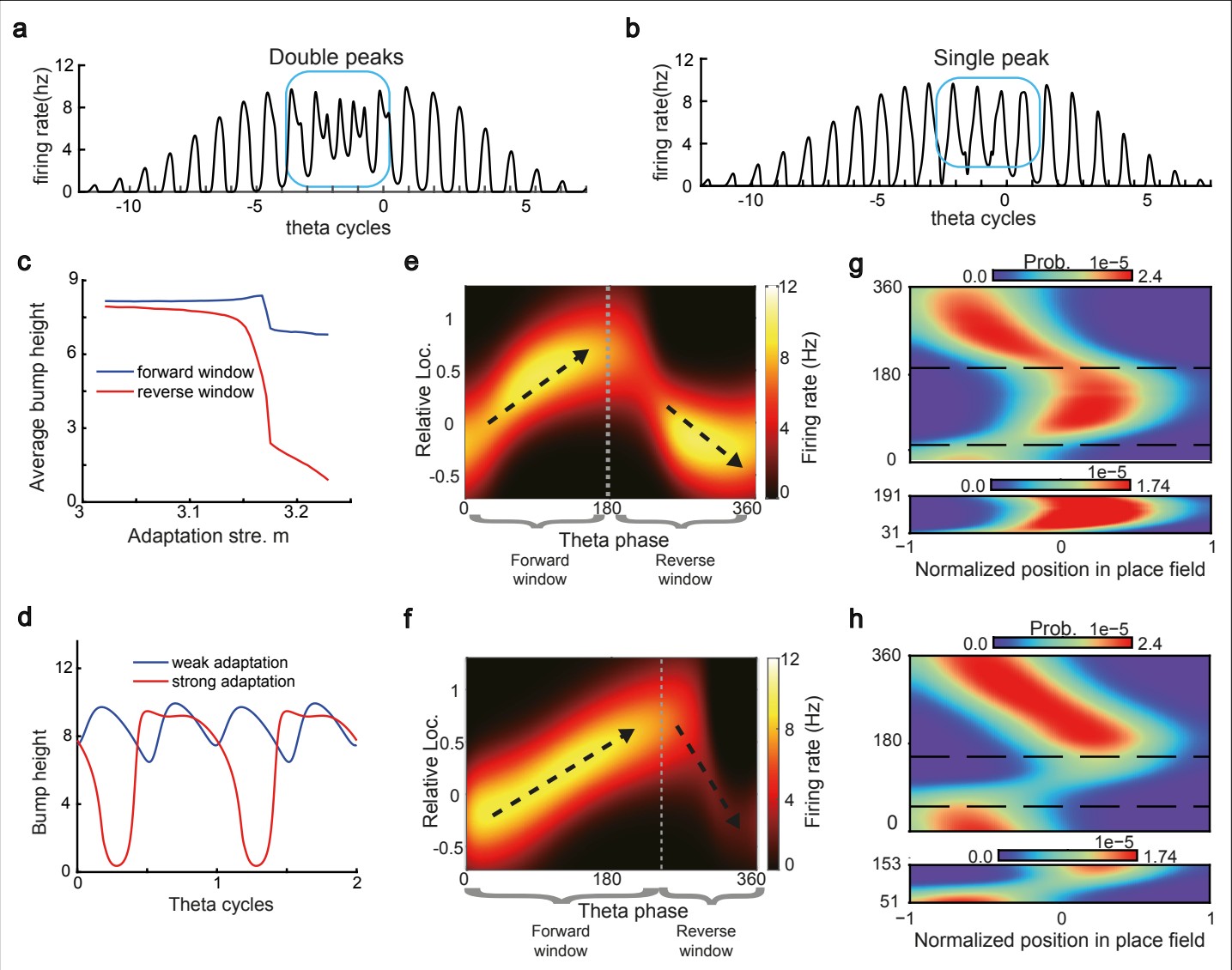

**Figure 4.** Different adaptation strengths account for the emergence of bimodal and unimodal cells. (**a**) The firing rate trace of a typical bimodal cell in our model. Blue boxes mark the phase shift stage. Note that there are two peaks in each theta cycle. (**b**) The firing rate trace of a typical unimodal cell. Note that there is only one firing peak in each theta cycle. For a comparison to (a) and (b), see experiment data shown in *Skaggs et al., 1996*, Figure 6. (**c**) The averaged bump heights in the forward (blue curve) and backward windows (red curve) as a function of the adaptation strength $m$. (**d**) Variation of the bump height when the adaptation strength is relatively small (blue line) or large (red line). (**e and f**) Relative location of the bump center in a theta cycle when adaptation strength is relatively small (**e**) or large (**f**). Dashed line separate the forward and backward windows. (**g and h**) Theta phase as a function of the normalized position of the animal in place field, averaged over all bimodal cells (**g**) or over all unimodal cells (**h**). −1 indicates that the animal just enters the place field, and 1 represents that the animal is about to leave the place field. Dashed lines separate the forward and backward windows. The lower panels in both (**g** and **h**) present the rescaled colormaps only in the backward window.

are asymmetric, i.e., forward sweeps always strongly suppress backward sweeps. On the contrary, the opposite effect is much smaller, since neuronal activities in backward sweeps have already been suppressed, and they can only generate weak adaptation. Because of this asymmetric suppression, the bump height in the forward sweep is always higher than that in the backward sweep (see *Figure 4c* and *Appendix 1—figure 3a*). When the adaptation strength $m$ is small, the suppression effect is not significant, and the attenuation of the bump height during the backward sweep is small (*Figure 4d*). In such case, the firing behavior of a place cell is similar to the situation as the bump height remains unchanged as analyzed in previous sections, i.e., the neuron can generate two firing peaks in a theta cycle at the phase shift stage, manifesting the property of a bimodal cell of having both significant

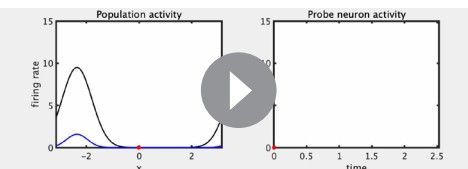

**Video 3.** The title of this video is: Neuronal activities during uni-directional oscillatory tracking state. https://elifesciences.org/articles/87055/figures#video3

phase precession and procession (*Figure 4e and g* and *Video 2*). When the adaptation strength *m* is large, the bump height in the backward sweep attenuates dramatically (see *Figure 4c and d* and the Video demonstration). As a result, the firing peak of a place cell in the backward sweep becomes nearly invisible at the phase shift stage, and the neuron exhibits only predominant phase precession, manifesting the property of a unimodal cell (*Figure 4f and h* and *Video 3*).

In summary, different adaptation strengths explain the emergence of bimodal and unimodal cells. In fact, there is no sharp separation between bimodal and unimodal cells. As the firing rate adaptation gets stronger, the network bump is more attenuated during the backward sweep, and cells with the bimodal firing property will gradually behave more like those with the unimodal firing property (see *Appendix 1—figure 3b*). Moreover, our model confirms that even though phase procession is weak, it still exists in unimodal cells (*Figure 4h*, lower panel), which has been reported in previous studies (*Wang et al., 2020*; *Yamaguchi et al., 2002*). This implies that phase procession is not a characteristic feature of bimodal cells, but instead, is likely a common feature of hippocampal activity, with a strength controlled by adaptation. Furthermore, the experimental data (*Fernández-Ruiz et al., 2017*) has indicated that there is a laminar difference between unimodal cells and bimodal cells, with bimodal cells correlating more with the firing patterns of deep CA1 neurons and unimodal cells with the firing patterns of superficial CA1 neurons. Our model suggests that this difference may come from the different adaptation strengths in the two layers.

## Constant cycling of multiple future scenarios in a T-maze environment

We have shown that our model can reproduce the forward and backward theta sweeps of decoded position when the animal runs on a linear track. It is noteworthy that there is only a single hypothetical future scenario in the linear track environment, i.e., ahead of the animal's position, and hence place cells firing phase can only encode future positions in one direction. However, flexible behaviors requires the animal encoding multiple hypothetical future scenarios in a quick and constant manner, e.g., during decision-making and planning in complex environments (*Johnson and Redish, 2007*; *Wikenheiser and Redish, 2015*). One recent study (*Kay et al., 2020*) showed constant cycling of theta sweeps in a T-maze environment (*Figure 5a*), i.e., as the animal approaches the choice point, the decoded position from hippocampal activity propagates down one of the two arms alternatively in successive LFP theta cycles. To reproduce this phenomenon, we change the structure of the CANN from a linear track shape to a T-maze shape where the neurons are aligned according to the location of their firing fields in the T-maze environment. Neurons are connected with a strength proportional to the Euclidean distance between their firing fields on the T-maze and the parameters are set such that the network is in the oscillatory tracking state (see details in 'Implementation details of the T-maze environment'). Mimicking the experimental protocol, we let the external input (the artificial animal) move from the end of the center arm to the choice point. At the beginning, when the external input is far away from the choice point, the network bump sweeps back and forth along the center arm, similar to the situation on the linear track. As the external input approaches the choice point, the network bump starts to sweep onto left and right arms alternatively in successive theta cycles (*Figure 5b* and *Video 4*; see also *Romani and Tsodyks, 2015*, for a similar model of cyclical sweeps spanning several theta cycles). The underlying mechanism is straightforward. Suppose that the bump first sweeps to the left arm from the current location, it will sweep back to the current location first due to the attraction of the external input. Then in the next round, the bump will sweep to the right arm, since the neurons on the left arm are suppressed due to adaptation. This cycling process repeats constantly between the two upcoming arms before the external input enters one of the two arms (i.e. before the decision is made). At the single cell level, this bump cycling phenomenon gives rise to the 'cycle skipping' effect (*Kay et al., 2020*; *Deshmukh et al., 2010*; *Brandon et al., 2013*), where a neuron whose place field is on one of the two arms fires on every other LFP theta cycle before the decision is made (*Figure 5c*, left panel and *Figure 5d*, upper panel). For example, a pair of cells with firing fields on each of the two

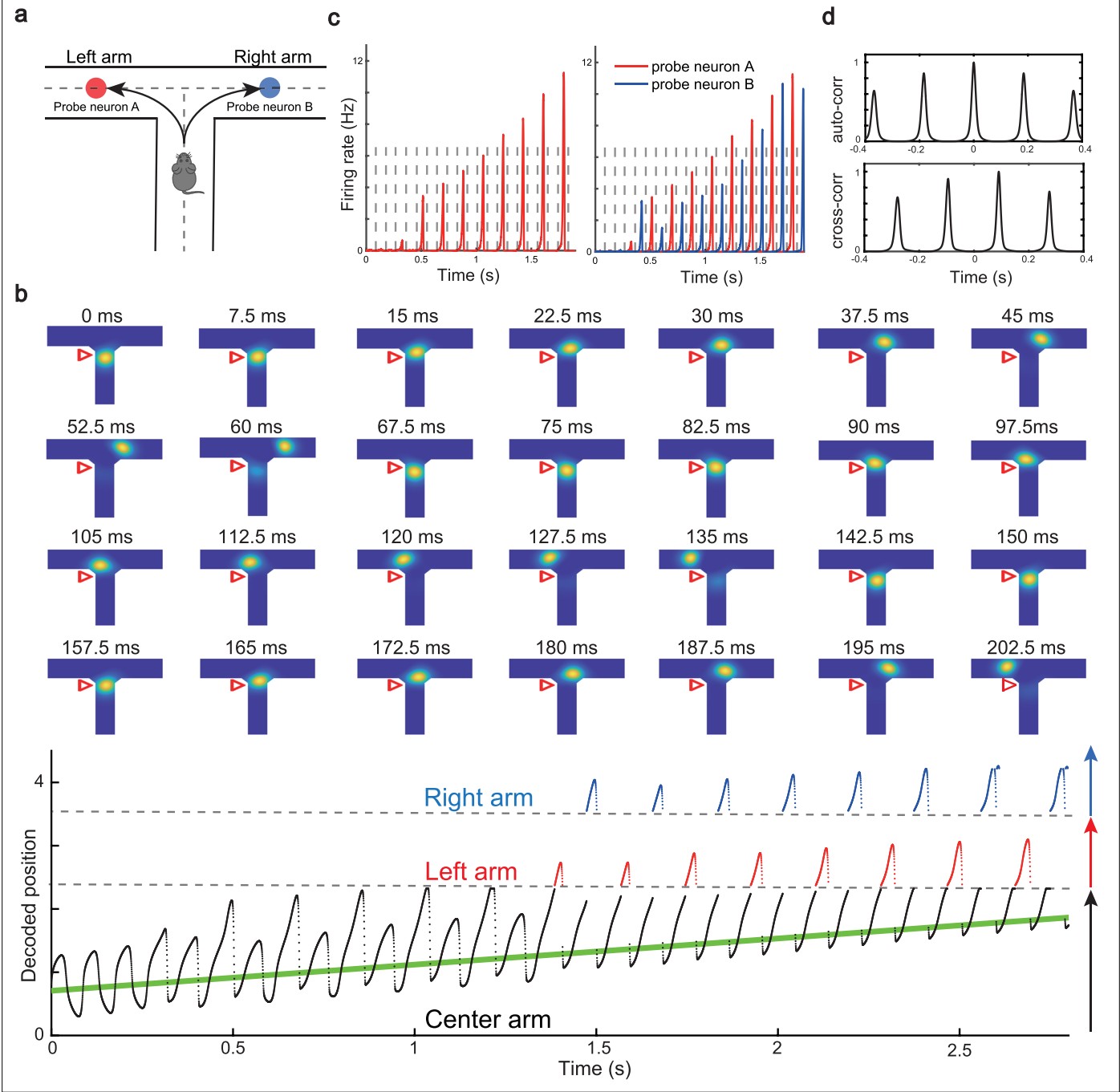

**Figure 5.** Constant cycling of future positions in a T-maze environment. (**a**) An illustration of an animal navigating a T-maze environment with two possible upcoming choices (the left and right arms). (**b**) Upper panel: Snapshots of constant cycling of theta sweeps on two arms when the animal is approaching the choice point. Red triangle marks the location of the external input. Note that the red triangle moves slightly toward the choice point in the 200 ms duration. Lower panel: Constant cycling of two possible future locations. The black, red, and blue traces represent the bump location on the center, left, and right arms, respectively. The green line marks the location of the external moving input. (**c**) Left panel: The firing rate trace of a neuron A on the left arm when the animal approaches the choice point. Right panel: The firing rate traces of a pair of neurons when the animal approaches the choice point, with neuron A (red) on the left arm and neuron B (blue) on the right arm. Dashed lines separate theta cycles. (**d**) Upper panel: The auto-correlogram of the firing rate trace of probe neuron A. Lower panel: The cross-correlogram between the firing rate trace of neuron A and the firing rate trace of neuron B.

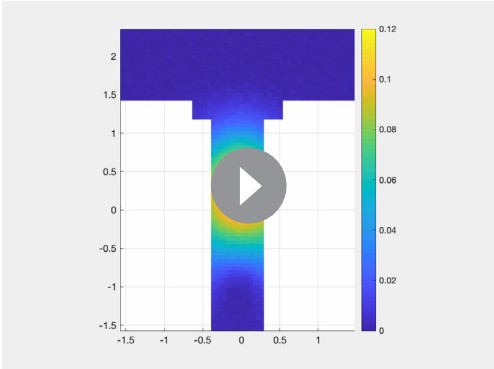

**Video 4.** The title of this video is: Bump oscillation in T-maze environment.
https://elifesciences.org/articles/87055/figures#video4

arms will fire in regular alternation on every other theta cycle (*Figure 5c*, right panel and *Figure 5d*, lower panel). These cell-level firing patterns agree well with the experimental observations (*Kay et al., 2020*).

In summary, our model, extended to a T-maze structure, explains the constant cycling of two possible future scenarios in a T-maze environment. The underlying mechanism relies on delayed adaptation, which alternately causes neurons on one arm to be more suppressed than those on the other arm. Such high-speed cycling may contribute to the quick and continuous sampling among multiple future scenarios in real-world decision-making and planning. We also note that there is a cyclical effect in the sweep lengths across oscillation cycles before the animal enters the left or right arm (see *Figure 5b*, lower panel), which may be interesting to check in the experimental data in future work (see Discussion for more details).

## Robust phase coding of position with place cells

As the firing rate shows large variability when the animal runs through the firing field (*Fenton and Muller, 1998*), it has been suggested that the theta phase shift provides an additional mechanism to improve the localization of animals (*O'Keefe and Burgess, 2005*). Indeed, *Jensen and Lisman, 2000*, showed that taking phase into account leads to a significant improvement in the accuracy of localizing the animal. To demonstrate the robustness of phase coding, previous experiments showed two intriguing findings: a linear relationship between the firing frequency of place cells and the animal's moving speed (*Geisler et al., 2007*), and the continued phase shift after interruption of hippocampal activity (*Zugaro et al., 2005*). We show that our model can also reproduce these two phenomena.

To investigate the relationship between the single cell's oscillation frequency and the moving speed as the animal runs through the firing field, we consider a unimodal cell with predominant phase precession as studied in *Geisler et al., 2007*. As we see from *Figure 3d* and *Figure 4a and b*, when the animal runs through the firing field of a place cell, the firing rate oscillates because the activity bump sweeps around the firing field center. Therefore, the firing frequency of a place cell has a baseline theta frequency, which is the same as the bump oscillation frequency. Furthermore, due to phase precession, there will be half a cycle more than the baseline theta cycles as the animal runs over the firing field, and hence single-cell oscillatory frequency will be higher than the baseline theta frequency (*Figure 6a*). The faster the animal runs, the faster the extra half cycle can be accomplished.

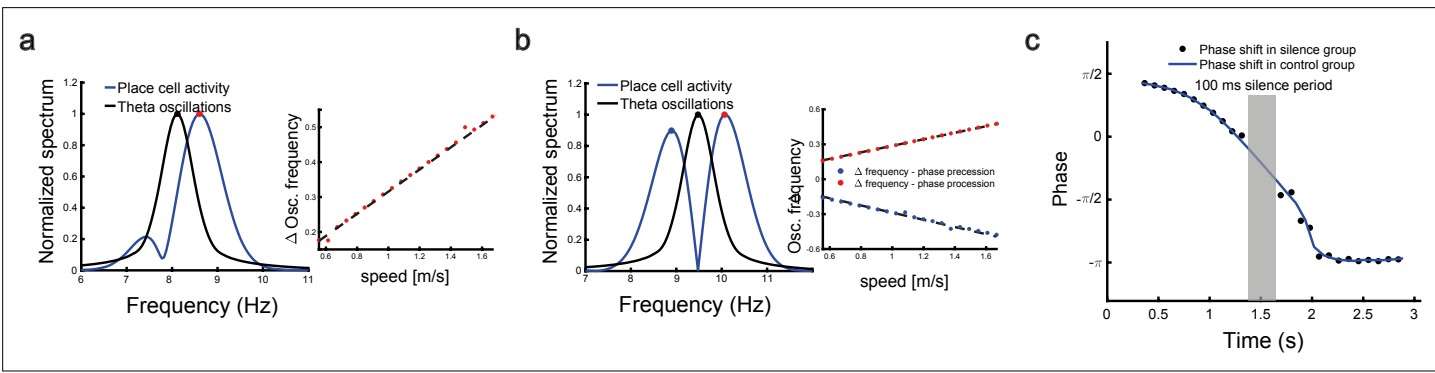

**Figure 6.** Robust phase coding of position. (**a**) Left: normalized spectrum of bump oscillation (black curve) and the oscillation of a unimodal cell (blue curve). Right: linear relationship between the frequency difference and the running speed. (**b**) Same as (a) but for a bimodal cell. (**c**) Silencing the network activity for 100 ms (grey shaded area) when the external moving input passes through the center part of the place field of a unimodal cell. Theta phase shifts of the unimodal cell are shown with (black points) or without (blue curve) silencing the network.

Consequently, the firing frequency will increase more (a steeper slope in *Figure 6a*, red dots) than the baseline frequency. This linear relationship ensures that the firing phase of a unimodal cell in each theta cycle is locked with the relative location of the animal in the firing field of that cell, which supports a robust phase-position code. Notably, in our model, the speed modulation of the place cells' firing frequency is not the cause of theta phase shift, but rather a result of oscillatory tracking. This is different from the dual oscillator model (*Lengyel et al., 2003*), which assumes that phase precession is caused by a speed-dependent increase in the dendritic oscillation frequency (see Discussion for more details).

In a different experiment, *Zugaro et al., 2005*, found that the firing phase of a place cell continues to precess even after hippocampal activity was transiently silenced for up to 250 ms (around 2 theta cycles). To reproduce this phenomenon, we also study a unimodal cell by manually turning off the network activity for a few hundred milliseconds (by setting $r(x,t) = 0$ for all neurons) and then letting the network dynamics evolves again with all parameters unchanged. Based on the theoretical analysis (*Equation 8*), we see that the firing phase is determined by the location of the animal in the place field, i.e., $v_{ext}t$. This means that the firing phase keeps tracking the animal's physical location. No matter how long the network is inactivated, the new firing phase will only be determined by the new location of the animal in the place field. Therefore, the firing phase in the first bump oscillation cycle after the network perturbation is more advanced than the firing phase in the last bump oscillation cycle right before the perturbation, and the amount of precession is similar to that in the case without perturbation (*Figure 6c*). This agrees well with the experimental observation (*Zugaro et al., 2005*), and indicates that the phase-position code is robust to the perturbation of the hippocampal dynamics.

Overall, our model reproduces these two experimental findings, and suggests that there exists a one-to-one correspondence between the firing phase of a place cell and the traveled distance in the neuron's place field, which is independent of the animal's running speed or the perturbation duration (*Appendix 1—figure 4*). This agrees well with experimental observations (*O'Keefe and Recce, 1993*) that theta phase correlates better with the animal's location than with time (*Figure 6*). In addition to the results for unimodal cells as introduced above, our model predicts new results for bimodal cells. First, in contrast to a unimodal cell, a bimodal cell will have two peaks in its firing frequency, with one slightly higher than the LFP theta baseline (due to phase precession) and the other slightly lower than the LFP theta baseline (due to phase procession). The precession-associated frequency positively correlates with the running speed of the animal, while the procession-associated frequency negatively correlates with the running speed (*Figure 6b*). Second, similar to the preserved phase shift in unimodal cells, both the phase precession and procession of a bimodal cell after transient intra-hippocampal perturbation continue from the new location of the animal (see *Appendix 1—figure 5*), no matter how long the silencing period lasts. The two predictions could be tested by experiments.

## Discussion
### Model contributions
In this paper, we have proposed a CANN with firing rate adaptation to unveil the underlying mechanism of place cell phase shift during locomotion. We show that the interplay between intrinsic mobility (owing to firing rate adaptation) and extrinsic mobility (owing to the location-dependent sensory inputs) leads to an oscillatory tracking state, which naturally accounts for theta sweeps where the decoded position oscillates around the animal's physical location at the theta rhythm. At the single neuron level, we show that the forward and backward bump sweeps account for, respectively, phase precession and phase procession. Furthermore, we show that the varied adaptation strength explains the emergence of bimodal and unimodal cells, i.e., as the adaptation strength increases, forward sweeps of the bump gradually suppress backward sweeps, and as a result, neurons initially exhibiting both significant phase precession and procession (due to a low-level adaptation) will gradually exhibit only predominant phase precession (due to a high-level adaptation).

### Computational models for theta phase shift and theta sweeps
As a subject of network dynamics, oscillatory tracking has been studied previously in an excitatory-inhibitory neural network (*Folias and Bressloff, 2004*), where it was found that decreasing the external input strength can lead to periodic emission of traveling waves in the network (Hopf instability), which

is analogous to the oscillatory tracking state in our model. However, their focus was on the mathematical analysis of such dynamical behavior, while our focus is on the biological implications of oscillatory tracking, i.e., how can it be linked to phase precession and procession of hippocampal place cells.

Due to their potential contributions to the temporal sequence learning involved in spatial navigation and episodic memory (*Mehta et al., 1997*; *Mehta et al., 2002*; *Yamaguchi, 2003*), theta phase precession and forward theta sweeps have been modeled in the field for decades. These models can be divided into two main categories, with one relying on the mechanism of single-cell oscillation (*O'Keefe and Recce, 1993*; *Kamondi et al., 1998*; *Lengyel et al., 2003*; *O'Keefe and Burgess, 2005*; *Mehta et al., 2002*) and the other relying on the mechanism of recurrent interactions between neurons (*Tsodyks et al., 1996*; *Romani and Tsodyks, 2015*; *Kang and DeWeese, 2019*). A representative example of the former is the oscillatory interference model (*O'Keefe and Recce, 1993*; *Lengyel et al., 2003*), which produces phase precession via the superposition of two oscillatory signals, with one from the baseline somatic oscillation at the LFP theta frequency (reflecting the inputs from the medial septal pacemaker; *Stewart and Fox, 1990*), and the other from the dendritic oscillation whose frequency is slightly higher. While these models can explain a large variety of experimental phenomena, it remain unclear how oscillation of individual neurons has a frequency higher than the baseline theta frequency. Here, our model provides a network mechanism for how such higher-frequency oscillation emerges.

A representative model relying on neuronal recurrent interactions is the activation spreading model (*Tsodyks et al., 1996*). This model produces phase precession via the propagation of neural activity along the movement direction, which relies on asymmetric synaptic connections. A later version of this model considers short-term synaptic plasticity (short-term depression) to implicitly implement asymmetric connections between place cells (*Romani and Tsodyks, 2015*), and reproduces many other interesting phenomena, such as phase precession in different environments. Different from these two models, our model considers firing rate adaptation to implement symmetry breaking and hence generates activity propagation. To prevent the activity bump from spreading away, their model considers an external theta input to reset the bump location at the end of each theta cycle, whereas our model generates an internal oscillatory state, where the activity bump travels back due to the attraction of external location input once it spreads too far away. Moreover, theoretical analysis of our model reveals how the adaptation strength affects the direction of theta sweeps, as well as offers a more detailed understanding of theta cycling in complex environments.

Based on our simulation, both STD and SFA show the ability to produce bi-directional sweeps within a CANN model, with the SFA uniquely enabling uni-directional sweeps in the absence of external theta inputs. This difference might be due to the lack of exhaustive exploration of the entire parameter space. However, it might also attribute to the subtle yet important theoretical distinctions between STD and SFA. Specifically, STD attenuates the neural activity through a reduction in recurrent connection strength, whereas SFA provides inhibitory input directly to the neurons, potentially impacting all excitatory inputs. These differences might explain the diverse dynamical behaviors observed in our simulations. Future experiments could clarify these distinctions by monitoring changes in synaptic strength and inhibitory channel activation during theta sweeps.

## Beyond the linear track environment

Besides the linear track environment, the mechanism of generating theta sweeps proposed in our model can also be generalized to more complex environments. For instance, in a T-maze environment, our model explains the constant cycling of theta sweeps between left and right arms. Such cycling behavior may be important for high-speed actions such as predating and escaping which require animals to make decision among several future scenarios at the sub-second level. Similar alternative activity sweeps in the T-maze environment has been studied in a previous paper (*Romani and Tsodyks, 2015*), which showed that the frequency of alternation correlates with overtly deliberative behaviors such as head scans (frequency at 1 Hz or less) (*Johnson and Redish, 2007*). In contrast to our model, the network activity in their model propagates continuously from the current location on the center arm till the end of the outer arm, which takes a few theta cycles (i.e. 1 s or more). In our model, the network bump alternately sweeps to one of the two outer arms at a much higher frequency (~ 8 Hz), which may be related to fast decision-making or planing in natural environments (*Kay et al., 2020*). Furthermore, our model can also be easily extended to the multiple-arms (>2)

environment (*Gillespie et al., 2021*) or the cascade-T environment (*Johnson and Redish, 2007*) with the underlying mechanism of generating theta cycling remaining unchanged. In addition to the linear and T-maze environments, phase shift has also been reported when an animal navigates in an open field environment. However, due to the lack of recorded neurons, decoding theta sweeps in the 2D environment is not as straightforward as in the 1D case. While theta sweeps in the 1D case have been associated with goal-directed behaviors and spatial planning (*Wikenheiser and Redish, 2015*), it remains unclear whether such conclusion is applicable to the 2D case. Our preliminary result shows that in the 2D CANN where neurons are arranged homogeneously according to their relative firing locations, the activity bump will sweep along the tangent direction of the movement trajectory, similar to the 1D case (see 'Oscillatory tracking in the 2D CANN – modeling theta sweeps in the open field environment' and *Appendix 1—figure 6* for details). It will be interesting to explore theta sweeps in the open field environment in detail when more experimental data is available.

## Model predictions and future works

Our model has several predictions which can be tested in future experiments. For instance, the height of the activity bump in the forward sweep window is higher than that in the backward sweep window (*Figure 4c*) due to the asymmetric suppression effect from the adaptation. For bimodal cells, they will have two peaks in their firing frequency as the animal runs across the firing fields, with one corresponding to phase precession and the other corresponding to phase procession. Similar to unimodal cells, both the phase precession and procession of a bimodal cell after transient intrahippocampal perturbation will continue from the new location of the animal (*Appendix 1—figure 7*). Interestingly, our model of the T-maze environment showed an expected phenomenon that as the animal runs toward the decision point, the theta sweep length also shows cyclical patterns (*Figure 5b*, lower panel). An intuitive explanation is that, due to the slow dynamics in firing rate adaptation (with a large time constant compared to neural firing), a long sweep leads to an adaptation effect on the neurons at the end of the sweep path. Consequently, the activity bump cannot travel as far due to the adaptation effect on those neurons, resulting in a shorter sweep length compared to the previous one. In the next round, the activity bump exhibits a longer sweep again because those neurons have recovered from the previous adaptation effect. We plan to test this phenomenon in future experiments.

In the current study, we have modeled the place cell population in the hippocampus with a CANN and adopted firing rate adaptation to generate theta phase shift. In fact, this model can be easily extended to the grid cell population without changing the underlying mechanism. For instance, we can induce the torus-like connection profile (periodic boundary in the 2D space) (*Samsonovich and McNaughton, 1997*; *McNaughton et al., 2006*) or the locally inhibitory connection profile (*Burak and Fiete, 2009*; *Couey et al., 2013*) in the CANN structure to construct a grid cell model, and by imposing firing rate adaptation, neurons in the grid cell network will also exhibit phase shift as the animal moves through the grid field, as reported in previous experimental studies (*Hafting et al., 2008*; *van der Meer and Redish, 2011*). Notably, although for both grid cells and place cells, CANNs can generate theta phase shift, it does not mean that they are independent from each other. Instead, they might be coordinated by the same external input from the environment, as well as by the medial septum which is known to be a pacemaker that synchronizes theta oscillations across different brain regions (*King et al., 1998*; *Wang, 2002*). We will investigate this issue in future work.

Our model also suggests that the 'online' theta sweep and the 'offline' replay may share some common features in their underlying mechanisms (*Romani and Tsodyks, 2015*; *Hopfield, 2010*; *Kang and DeWeese, 2019*; *Jahnke et al., 2015*). We have shown that the activity bump with strong adaptation can move spontaneously when the external input becomes weak enough (see previous sections). Such non-local spreading of neural activity has a speed much faster than the conventional speed of animals (the external input speed in our model, see *Figure 2d*), which resembles the fast spreading of the decoded position during sharp wave-ripple events (*Diba and Buzsáki, 2007*; *Foster and Wilson, 2006*; *Karlsson and Frank, 2009*; *Dragoi and Tonegawa, 2011*). This indicates that these two phenomena may be generated by the same neural mechanism of firing rate adaptation, with theta sweeps originating from the interplay between the adaptation and the external input, while replay originating from only the adaptation, since the external input is relatively weak during the 'offline' state. This hypothesis seems to be supported by the coordinated emergence of theta sequences and replays during the post-natal development period (*Muessig et al., 2019*), as well as

their simultaneous degradation when the animal traveled passively on a model train (*Drieu et al., 2018*).

Nevertheless, it is important to note that the CANN we adopt in the current study is an idealized model for the place cell population, where many biological details are missed (*Amari, 1977*; *Tsodyks and Sejnowski, 1995*; *Samsonovich and McNaughton, 1997*; *Tsodyks, 1999*). For instance, we have assumed that neuronal synaptic connections are translation-invariant in the space. In practice, such a connection pattern may be learned by a synaptic plasticity rule at the behavioral timescale when the animal navigates actively in the environment (*Bittner et al., 2017*). In future work, we will explore the detailed implementation of this connection pattern, as well as other biological correspondences of our idealized model, to establish a comprehensive picture of how theta phase shift is generated in the brain.

## Materials and methods
### General summary of the model

We consider a 1D CANN, in which neurons are uniformly aligned according to their firing fields on a linear track (for the T-maze case, see 'Implementation details of the T-maze environment' below; for the case of the open field (2D CANN), see 'Oscillatory tracking in the 2D CANN – modeling theta sweeps in the open field environment'). Denote $U(x,t)$ the synaptic input received by the place cell at location $x$, and $r(x,t)$ the corresponding firing rate. The dynamics of the network is written as:

$$\tau \frac{dU(x,t)}{dt} = -U(x,t) + \rho \int_{-\infty}^{\infty} J(x,x')r(x',t)dx' - V(x,t) + I^{ext}(x,t),\tag{9}$$

where $\tau$ is the time constant of $U(x,t)$ and $\rho$ the neuron density. The firing rate $r(x,t)$ is given by:

$$r(x,t) = \frac{gU(x,t)^2}{1 + k\rho \int_{-\infty}^{\infty} U(x',t)^2\, dx'},\tag{10}$$

where $k$ controls the strength of the global inhibition (divisive normalization), $g$ denotes a gain factor. $J(x,x')$ denotes the connection weight between place cells at location $x$ and $x'$, which is written as:

$$J(x,x') = \frac{J_0}{\sqrt{2\pi}a} exp\left[-\frac{(x-x')^2}{2a^2}\right],\tag{11}$$

where $J_0$ controls the strength of the recurrent connection and $a$ the range of neuronal interaction. Notably, $J(x,x')$ depends on the relative distance between two neurons, rather than the absolute locations of neurons. Such translation-invariant connection form is crucial for the neutral stability of the attractor states of CANNs (*Wu et al., 2016*). $I^{ext}(x,t)$ represents the external input which conveys the animal location information to the hippocampal network, which is written as:

$$I^{ext}(x,t) = \alpha exp\left[-\frac{(x-v_{ext}t)^2}{4\sigma^2}\right],\tag{12}$$

with $v_{ext}$ denoting the animal's running speed and $\alpha$ controlling the input strength to the hippocampus. $\sigma$ denotes the width of the external input $I^{ext}$, which is set to be equal to the recurrent connection width $a$ in the main text and the following derivation. $V(x,t)$ denotes the adaptation effect of the place cell at location $x$, which increases with the synaptic input (and hence the place cell's firing rate), i.e.,

$$\tau_v \frac{dV(x,t)}{dt} = -V(x,t) + mU(x,t),\tag{13}$$

with $\tau_v$ denoting the time constant of $V(x,t)$ and $m$ the adaptation strength. Note that $\tau_v \gg \tau$, meaning that adaptation is a much slower process compared to the neural firing.

## Stability analysis of the bump state

We derive the condition under which the bump activity is the stable state of the CANN. For simplicity, we consider the simplest case that there is no external input and adaptation in the network, i.e., $m = \alpha = 0$. In this case, the network state is determined by the strength of the recurrent excitation and global inhibition. When the global inhibition is strong ($k$ is large), the network is silent, i.e., no bump activity emerges in the CANN. When the global inhibition is small, an activity bump with the Gaussian-shaped profile emerges, which is written as:

$$\overline{U}(x, t) = A_u exp\left\{ -\frac{[x - z(t)]^2}{4a^2} \right\},$$ (14)

$$\overline{r}(x, t) = A_r exp\left\{ -\frac{[x - z(t)]^2}{2a^2} \right\},$$ (15)

with $A_u$ and $A_r$ representing the amplitudes of the synaptic input bump and the firing rate bump, respectively. $z(t)$ represents the bump center, and $a$ is the range of neuronal interaction (defined in 'General summary of the model'). To solve the network dynamics, we substitute **Equations 14 and 15** into **Equations 9 and 10**, which gives (see 'Deriving the network state when the external input does not exist ($I^{ext} = 0$)' for more details of the derivation):

$$\tau \frac{dA_u}{dt} = -A_u + \frac{\rho J_0}{\sqrt{2}} A_r,$$ (16)

$$A_r = \frac{A_u^2}{1 + \sqrt{2\pi} k \rho a A_u^2},$$ (17)

These two equations describe how the bump amplitudes change with time. For instance, if neurons are weakly connected (small $J_0$) or they are connected sparsely (small $\rho$), the second term on the right-hand side of **Equation 16** is small, and $A_u$ will decay to zero, implying that the CANN cannot sustain a bump activity. By setting $dA_u/dt = 0$, we obtain:

$$A_u = \frac{\rho J_0 \pm \sqrt{\rho^2 J_0^2 - 8\sqrt{2\pi} 2k \rho a}}{4\sqrt{\pi} k \rho a},$$ (18)

$$A_r = \frac{\sqrt{2}}{\rho J_0} A_u.$$ (19)

It is straightforward to check that only when:

$$k < k_c = \rho J_0^2 / 8\sqrt{2\pi} a,$$ (20)

$A_u$ have two real solutions (indicated by the ± sign in **Equation 18**), i.e., the dynamic system (**Equations 16 and 17**) has two fixed points. It can be checked that only $A_u = \left( \rho J_0 + \sqrt{\rho^2 J_0^2 - 8\sqrt{2\pi} 2k \rho a} \right) / \left( 4\sqrt{\pi} k \rho a \right)$ is the stable solution.

## Analysis of the intrinsic mobility of the bump state

We derive the condition under which the bump of the CANN moves spontaneously in the attractor space without relying on external inputs. As the adaptation strength increases, the bump activity becomes unstable and has tendency to move away from its location spontaneously. Such intrinsic mobility of the CANN has been shown in previous studies (**Bressloff, 2012**; **Wu et al., 2016**; **Mi et al., 2014**). We set $\alpha = 0$ (no external input), and investigate the effect of adaptation strength $m$ on the bump dynamics. Our simulation result shows that during the spontaneous movement, $V(x, t)$ can also be represented by a Gaussian-shaped bump, which is written as:

$$\overline{V}(x, t) = A_v exp\left\{ -\frac{[x - z(t) + d(t)]^2}{4a^2} \right\},$$ (21)

where $A_v$ denotes the amplitude of the adaptation bump, and $d(t)$ the displacement between the bump centers of $U(x,t)$ and $V(x,t)$. This displacement originates from the slow dynamics of adaptation, which leads to the adaptation bump always lags behind the neural activity bump. Similar to 'Stability analysis of the bump state', we substitute the bump profiles *Equations 14, 15, 21* into the network dynamics *Equations 9, 10, 13*, and obtain:

$$\tau \left[ A_u \frac{x-z}{2a^2} \frac{dz}{dt} + \frac{dA_u}{dt} \right] \mathcal{N}(x,z,2a) = \left( -A_u + \frac{\rho J_0}{\sqrt{2}} A_r \right) \mathcal{N}(x,z,2a) - A_v \mathcal{N}(x,z-d,2a), \tag{22}$$

$$A_r = \frac{A_u^2}{1 + k\rho\sqrt{2\pi}aA_u^2}, \tag{23}$$

$$\tau_v \left[ A_v \frac{x-z+d}{2a^2} \frac{d(z-d)}{dt} + \frac{dA_v}{dt} \right] \mathcal{N}(x,z-d,2a) = -A_v \mathcal{N}(x,z-d,2a) + mA_u \mathcal{N}(x,z,2a), \tag{24}$$

where $N(x,z,2a) = exp\left\{ -[x-z]^2/4a^2 \right\}$.

At first glance, the resulting equations given by *Equations 22 and 24* may seem intractable due to the high dimensionality (i.e. $2N$, where $N$ is the number of neurons in the network). However, a key property of CANNs is that their dynamics are dominated by a few motion modes, which correspond to distortions of the bump shape in terms of height, position, width, etc. (*Fung et al., 2010*). By projecting the network dynamics onto its dominant motion modes (*Fung et al., 2010*) (which involves computing the inner product of a function $f(x)$ with a mode $u_n(x)$), we can significantly simplify the network dynamics. Typically, projecting onto the first two motion modes is sufficient to capture the main features of the dynamics, which are given by:

$$u_0(x,t) = exp\left\{ -\frac{[x-z(t)]^2}{4a^2} \right\}, \tag{25}$$

$$u_1(x,t) = [x-z(t)] exp\left\{ -\frac{[x-z(t)]^2}{4a^2} \right\}. \tag{26}$$

By projecting the network dynamics onto these two motion modes, we obtain:

$$-A_u + \frac{\rho J_0}{\sqrt{2}} A_r - A_v exp\left( -\frac{d^2}{8a^2} \right) = 0, \tag{27}$$

$$\tau A_u v_{int} = dA_v exp\left( -\frac{d^2}{8a^2} \right), \tag{28}$$

$$\frac{d}{4a^2} \tau_v A_v exp\left( -\frac{d^2}{8a^2} \right) v_{int} = -A_v exp\left( -\frac{d^2}{8a^2} \right) + mA_u, \tag{29}$$

$$\tau_v \left( 1 - \frac{d^2}{4a^2} \right) v_{int} = d. \tag{30}$$

Note that we assume that the bump height keep as constant over time, i.e., $dA_u/dt = dA_v/dt = 0$ is assumed. *Equations 27–30* describe the relationships between bump features $A_u, A_r, A_v, v_{int}$, and $d$, where $v_{int} = dz/dt$ representing the intrinsic moving speed of the bump center. By solving these equations together with *Equation 23*, we obtain:

$$A_u = \frac{\rho J_0 + \sqrt{\rho^2 J_0^2 - 8\sqrt{2\pi}k\rho a \left( 1 + \sqrt{\frac{m\tau}{\tau_v}} \right)^2}}{4\sqrt{\pi}k\rho a \left( 1 + \sqrt{\frac{m\tau}{\tau_v}} \right)}, \tag{31}$$

$$A_r = \frac{\rho J_0 + \sqrt{\rho^2 J_0^2 - 8\sqrt{2\pi}k\rho a\left(1 + \sqrt{\frac{m\tau}{\tau_v}}\right)^2}}{2\sqrt{2\pi}k\rho^2 a J_0},$$ (32)

$$A_v = \sqrt{\frac{m\tau}{\tau_v}}exp\left[\frac{1 - \sqrt{\frac{\tau}{m\tau_v}}}{2}\right]\frac{\rho J_0 + \sqrt{\rho^2 J_0^2 - 8\sqrt{2\pi}k\rho a\left(1 + \sqrt{\frac{m\tau}{\tau_v}}\right)^2}}{4\sqrt{\pi}k\rho a\left(1 + \sqrt{\frac{m\tau}{\tau_v}}\right)},$$ (33)

$$d = 2a\sqrt{1 - \sqrt{\frac{\tau}{m\tau_v}}},$$ (34)

$$v_{int} = \frac{2a}{\tau_v}\sqrt{\frac{m\tau_v}{\tau} - \sqrt{\frac{m\tau_v}{\tau}}}.$$ (35)

*Equations 31–33* describe the amplitudes of the bumps of synaptic input, firing rate, and adaptation in the CANN, respectively, and *Equation 34* describes the displacement between the neural activity and adaptation bumps. From *Equation 35*, we see that for the bump to travel spontaneously, it requires $m > \tau/\tau_v$, i.e., the adaptation strength is larger than a threshold given by the ratio between two time constants $\tau$ and $\tau_v$. As the adaptation strength increases (larger $m$), the traveling speed of the bump increases (larger $v_{int}$).

## Analysis of the oscillatory tracking behavior of the bump state

When both the external input and the adaptation are applied to the CANN, the bump activity can oscillate around the external input if the strengths of the external input and the adaptation are appropriated. The simulation shows that during the oscillatory tracking, the bump shape is roughly unchanged, and the oscillation of the bump center can be approximated as a sinusoidal wave expressed as:

$$z(t) = c_0 sin(\omega t) + d_0 + v_{ext}t,$$ (36)

where $c_0$ and $\omega$ denote, respectively, the oscillation amplitude and frequency, and $d_0$ denotes a constant offset between the oscillation center and the external input.

Similar to the analysis in 'Analysis of the intrinsic mobility of the bump state', we substitute the expression of $z(t)$ (*Equation 36*) into *Equations 14, 15, 21*, and then simplify the network dynamics by applying the projection method (see 'Deriving the oscillatory tracking state of the network when the external input is applied ($I^{ext} \neq 0$)' for more detailed derivation). We obtain:

$$(m + 1)A_u - \frac{\rho J_0}{\sqrt{2}}\frac{A_u^2}{1 + \sqrt{2\pi}ak\rho A_u^2} - \alpha = 0,$$ (37)

$$\omega^2 = \frac{\alpha}{\tau\tau_v A_u},$$ (38)

$$mA_u exp\left(-\frac{d^2}{8a^2}\right) = A_v,$$ (39)

$$d_0 = \tau_v v,$$ (40)

$$\sqrt{\frac{2(\tau A_u + \alpha\tau_v)}{\alpha\tau_v}\left[4a^2\left(ln\frac{\tau_v m A_u}{\tau A_u + \alpha\tau_v}\right) - \tau_v^2 v^2\right]} = c_0,$$ (41)

*Equations 37–41* describe the relationships among six oscillation features $A_u, A_r, A_v, c_0, d_0$, and $\omega$. By solving these equations, we obtain:

$$A_u = \frac{J_0 + 2\sqrt{\pi}ak\alpha}{2\sqrt{\pi}ak(1 + m)},$$ (42)

$$A_r = \frac{A_u^2}{1 + \sqrt{2\pi}ak\rho A_u^2},$$ (43)

$$A_v = \sqrt{\left(\frac{\tau A_u + \alpha \tau_v}{\tau_v}\right) m A_u},$$ (44)

$$c_0 = A_v \sqrt{\frac{2}{\alpha m A_u} \left[8a^2 ln \frac{m A_u}{A_v} - \tau_v^2 v^2\right]},$$ (45)

$$d_0 = \tau_v v,$$ (46)

$$\omega = \sqrt{\frac{\alpha}{\tau \tau_v A_u}}.$$ (47)

It can be seen from **Equation 45** that for the bump activity to oscillate around the external input (i.e. the oscillation amplitude $c_0 > 0$), it requires that $8a^2 ln (m A_u/A_v) - \tau_v^2 v^2 > 0$. This condition gives the boundary (on the parameter values of the input strength $\alpha$ and the adaptation strength $m$) that separate two tracking states, i.e., smooth tracking and oscillatory tracking (see **Figure 2g** and **Appendix 1—figure 8** for the comparison between the simulation results and theoretical results).

Note that to get the results in **Equations 37–41**, we have assumed that the amplitudes of neural activity bumps and the adaptation bump remain unchanged during the oscillation (i.e. $A_u, A_v, A_r$ are constants). However, this assumption is not satisfied when the SFA strength $m$ is large (see previous sections and **Figure 4**). In such a case, we carry out simulation to analyze the network dynamics.

## Implementation details of the linear track environment

For the linear track environment, we simulate an 1D CANN with 512 place cells topographically organized on the 1D neuronal track. Since we are interested in how the neuronal firing phase shifts as the animal moves through the firing field of a place cell, we investigate the place cell at location $x = 0$ and ignore the boundary effect, i.e., we treat the linear track with the infinite length. The neural firing time constant is set to be 3 ms, while the time constant of spike frequency adaptation is much longer, which is set to be 144 ms. The density of place cells on the linear track is set to be $256/\pi$. The excitatory interaction range of place cells is set to be $0.4m$, while the maximum excitatory connection strength $J_0$ is set to be 0.2. The gain factor is set to be 5. The global inhibition strength $k$ is set to be 5. The moving speed of the virtual animal $v_{ext}$ is set to be 1.5 m/s. For the simulation details, we use the first-order Euler method with the time step $\delta t$ set to be 0.3, the duration of simulation $T$ set to be 10 s. These parameters are commonly used in all plots related to the linear track environment (see **Appendix 1—table 1** for a summary).

For the two key parameters, i.e., the external input strength $\alpha$ and the adaptation strength $m$, we vary their values in different plots. Specifically, for illustrating the smooth tracking state in **Figure 2c**, we set $\alpha = 0.19$ and $m = 0$. For illustrating the traveling wave state (intrinsic mobility of the bump state) in **Figure 2d**, we set $\alpha = 0$ and $m = 0.31$. For plotting the relationship between the intrinsic speed $v_{int}$ and the adaptation strength $m$ shown in **Figure 2e**, we keep $\alpha = 0$, but vary $m$ in the range between 0 and 0.1 with a step of 0.05. For plotting the overall phase diagram including all three moving states as shown in **Figure 2g**, we vary $\alpha$ in the range between 0.05 and 0.16 with a step of 0.001, and $m$ in the range between 0.9 and 1.8 with a step of 0.01. To generate bimodal cell firing patterns in **Figure 3a** and **Figure 4a, e, and g**, we choose $\alpha = 0.19$ and $m = 3.02$. To generate unimodal firing patterns in **Figure 4b, f, and h**, we choose $\alpha = 0.19$ but a relatively larger adaptation strength with $m = 3.125$. The values of these two parameters in different plots are summarized in **Appendix 1—table 2**.

## Implementation details of the T-maze environment

### Parameter configurations during simulation

To simulate the T-maze environment, we consider a CANN in which place cells are topographically organized in a T-shaped area which consists of a vertical central arm and two horizontal left and right arms (**Figure 5a**). The width of the central arm is set to be 0.84 m and the length is set to be 3.14 m. The widths of the two horizontal arms are also set to be 0.84 m, while the lengths of both arms are set to be 2.36 m. The connection strength between two neurons is determined by the distance between them, which is written as:

$$J(x, x'; y, y') = \frac{J_0}{2\pi a^2} exp\left[-\frac{(x-x')^2 + (y-y')^2}{2a^2}\right].$$ (48)

Here, $(x, y)$ and $(x', y')$ represent the coordinates of two neurons in the T-maze environment, $a$ is the recurrent connection range which is set to be 0.3, and $J_0$ controls the connection strength which is set to be 0.0125. Since we are interested in investigating theta sweeps when the animal is running on the central arm toward the junction point, the external input is restricted on the central arm which is modeled by a Gaussian-like moving bump written as:

$$I^{ext}(x, y) = \alpha exp\left[-\frac{(x - x_0)^2 + (y - y_0)^2}{2a^2}\right],$$  (49)

where $x_0 = 0$ and $y_0 = v_{ext}t$ represent the center location of the external input with a moving speed $v_{ext} = 1.5$ m/s. In the simulation, we used the first-order Euler method with the time step $\delta t = 0.3$ s and the duration of simulation $T = 4.2$ s. The parameters used are summarized in *Appendix 1—table 3*.

## Calculating auto-correlogram and cross-correlogram

To show the 'cycle skipping' effect of a single place cell in the T-maze environment, we calculate the auto-correlogram of the firing rate trace of a place cell whose firing field encodes a location on the left arm (the upper panel in *Figure 5d*). Assume the firing trace of the place cell is $f(t)$ (shown in left panel in *Figure 5c*), the auto-correlogram is calculated as:

$$(f * f)(\tau) \triangleq \int_{-\infty}^{\infty} f(t) f(t + \tau)\, dt,$$  (50)

where $\tau$ represents the time offset.

To show the 'alternative cycling' effect of a pair of place cells with each of them encoding a location on each of the two outward arms, we calculate the cross-correlogram between their firing traces (the lower panel in *Figure 5d*). It measures the similarity of the two firing traces as a function of the temporal offset of one relative to the other. Assume the firing traces of the two place cells are $f(t)$ and $g(t)$, respectively, the cross-correlogram is calculated as:

$$(f * g)(\tau) \triangleq \int_{-\infty}^{\infty} f(t) g(t + \tau)\, dt,$$  (51)

where $\tau$ represents the time offset.

## Details of generating the probability heatmap of theta phase shift

In *Figure 4g and h* we described the smoothed probability heatmaps of theta phase versus normalized position in the place field of both bimodal and unimodal cells. Generally, these two plots are similar to the traditional spike plot of phase and position traveled in the place field (*O'Keefe and Recce, 1993*; *Skaggs et al., 1996*). However, in our rate-based model, the phase of neuronal spike is not directly modeled, rather we use the phase of firing rate peak to represent the phase shift in neuronal firing. Here, we describe the implementation details of generating the heatmaps.

The *x*-axis denotes the normalized position in the place field, with –1 representing the position where the animal just enters the place field, and 1 representing the position where the animal just leaves the place field. In our simulation, the firing field of a place cell with preferred location at $x_0$ is defined as $x \in (x_0 - 2.5 * a, x_0 + 2.5a)$, with $a$ roughly the half size of the firing field. Consider the animal is at $x_t$ at time $t$ (note that $x_t = v_{ext}t$), then its normalized position $x_t$ is calculated as $x_t = (x_t - x_0) / (5a)$. The *y*-axis represents the phase of neuronal activity, which is in the range of (0°, 720°). To calculate the phase at every time step, we divide the duration of the animal traversing the linear track into multiple theta cycles according to the bump's oscillation. We can calculate the phase by $\theta_t = (t - t_0) / T$, with $t_0$ referring to the beginning of the present theta cycle and $T$ referring to the theta period. Denote the firing rate of the *i*th neuron at time $t$ as $r_i(x_t, \theta_t)$, the probability heatmap is calculated by:

$$p(x, \theta_t) = C \sum_{i=1}^{N_c} \theta_t r_i(x, \theta_t),$$  (52)

where $C = 1/\sum_t \sum_{i=1}^{N_c} \theta_t r_i(x, \theta_t)$ is the normalization factor.

## Spike generation from the firing rate

To understand phase shift based on spiking time rather than the peak firing rate, we convert the firing rate into spike trains according to the Poisson statistics (note that our analysis is rate-based, but converting to spike-based does not change the underlying mechanism). For the $i$th place cell which encodes position $x_i$ on the linear track, the number of spikes $n_i$ it generates within a time interval $\Delta t$ satisfies a Poisson distribution, which is expressed as:

$$P\left(n_i|z\right) = \frac{\left[f_i\left(z\right)\Delta t\right]^{n_i}}{n_i!} e^{-f_i(z)\Delta t}, \tag{53}$$

where $z$ is the animal's location, and $f_i\left(z\right)$ is the tuning function of cell $i$, which is given by:

$$f_i\left(z\right) = A_r exp\left[-\frac{\left(x_i - z\right)^2}{2a^2}\right], \tag{54}$$

where $A_r$ denotes the amplitude of the neural activity bump and $a$ the range of recurrent interaction.

## Acknowledgements

We thank Brad Pfeiffer for sharing the data. We also thank Brad Pheiffer, Cheng Wang, Li Yao for valuable discussions. This work was support by: Science and Technology Innovation 2030-Brain Science and Brain-inspired Intelligence Project (No.2021ZD0200204, SW; No.2021ZD0200204/2021ZD0203700/2021ZD0203705, YM), the UKRI Frontier Research Grant (EP/X023060/1, DB), the Wellcome Principal Research Fellowship (NB), the National Natural Science Foundation of China (No. 62088102/62136001/T2122016, YM), and an International Postdoctoral Exchange Fellowship Program (No. PC2021005, ZJ).

## Additional information

### Funding

| Funder | Grant reference number | Author |
|---|---|---|
| Science and Technology Innovation 2030-Brain Science and Brain-inspired Intelligence Project | 2021ZD0200204 | Si Wu<br>Yuanyuan Mi |
| Science and Technology Innovation 2030-Brain Science and Brain-inspired Intelligence Project | 2021ZD0203700 | Yuanyuan Mi |
| Science and Technology Innovation 2030-Brain Science and Brain-inspired Intelligence Project | 2021ZD0203705 | Yuanyuan Mi |
| UK Research and Innovation | Frontier Research Grant EP/X023060/1 | Daniel Bush |
| Wellcome | Principal Research Fellowship | Neil Burgess |
| National Natural Science Foundation of China | 62088102 | Yuanyuan Mi |
| National Natural Science Foundation of China | 62136001 | Yuanyuan Mi |
| National Natural Science Foundation of China | T2122016 | Yuanyuan Mi |

| Funder | Grant reference number | Author |
|--------|------------------------|--------|
| International Postdoctoral Exchange Fellowship Program | PC2021005) | Zilong Ji |

The funders had no role in study design, data collection and interpretation, or the decision to submit the work for publication. For the purpose of Open Access, the authors have applied a CC BY public copyright license to any Author Accepted Manuscript version arising from this submission.

## Author contributions

Tianhao Chu, Conceptualization, Formal analysis, Investigation, Visualization, Writing - original draft, Writing - review and editing; Zilong Ji, Conceptualization, Formal analysis, Funding acquisition, Investigation, Visualization, Writing - original draft, Writing - review and editing; Junfeng Zuo, Formal analysis, Visualization; Yuanyuan Mi, Tiejun Huang, Supervision; Wen-hao Zhang, Conceptualization; Daniel Bush, Investigation; Neil Burgess, Supervision, Investigation; Si Wu, Conceptualization, Supervision, Writing - original draft, Writing - review and editing

## Author ORCIDs

Tianhao Chu ⓘ http://orcid.org/0000-0001-9910-9361
Zilong Ji ⓘ http://orcid.org/0000-0001-7868-6178
Daniel Bush ⓘ http://orcid.org/0000-0002-5097-8117
Neil Burgess ⓘ http://orcid.org/0000-0003-0646-6584
Si Wu ⓘ http://orcid.org/0000-0001-9650-6935

Reviewer #1 (Public review): https://doi.org/10.7554/eLife.87055.4.sa1
Reviewer #2 (Public review): https://doi.org/10.7554/eLife.87055.4.sa2
Author response https://doi.org/10.7554/eLife.87055.4.sa3

# Additional files

## Supplementary files

- MDAR checklist

- Source code 1. Modelling code for 'Firing rate adaptation affords place cell theta sweeps, phase precession, and procession'.

## Data availability

The current manuscript is a computational study, so no data have been generated for this manuscript. Modelling code has been uploaded as *Source code 1*.

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

# Appendix 1

## 1 The network model

We consider a CANN, in which neurons are uniformly distributed in a 1D environment, mimicking place cells rearranged according to the locations of their firing fields on the linear track. Neurons in the CANN are connected with each other recurrently. Denote $U(x, t)$ the synaptic input received by neurons at location $x$, with $x \in (-\infty, \infty)$, and $r(x, t)$ the corresponding firing rate. The dynamics of the network are written as:

$$\tau \frac{dU(x,t)}{dt} = -U(x,t) + \rho \int_{-\infty}^{\infty} J(x,x')r(x',t)dx' - V(x,t) + I^{ext}(x,t), \tag{A1}$$

$$\tau_v \frac{dV(x,t)}{dt} = -V(x,t) + mU(x,t), \tag{A2}$$

$$r(x,t) = \frac{U(x,t)^2}{1 + k\rho \int_{-\infty}^{\infty} U(x',t)\,dx'}, \tag{A3}$$

where $\tau$ is the time constant of $U(x, t)$, $\rho$ the neuron density, and $I^{ext}(x, t)$ the external input. $J(x, x')$ represents the connection strength between neurons at $x$ and $x'$. $V(x, t)$ representing the effect of firing rate adaptation on this neuron. $\tau_v$ is the time constant of $V(x, t)$, and $m$ controls the adaptation strength. The parameter $k$ controls the amount of divisive normalization, reflecting the contribution of inhibitory neurons (not explicitly modeled) (*Mitchell and Silver, 2003*).

The connection profile between two neurons with firing fields at location $x$ and $x'$ is set as:

$$J(x,x') = \frac{J_0}{2\pi a} exp\left[-\frac{(x-x')^2}{2a^2}\right], \tag{A4}$$

where the parameter $J_0$ controls the amount of recurrent connection strength and $a$ represents the range of neuronal interactions. Notably, the recurrent connection between place cells is translation-invariant, i.e., $J(x, x')$ is a function of $(x - x')$. This feature is crucial for the neutral stability of continuous attractor networks.

It is known that when there is no external input or firing rate adaptation ($m = 0, I^{ext} = 0$), the CANN can hold a continuous family of Gaussian-shaped stationary states (*Fung et al., 2010*; *Fung et al., 2012*), called bump activities, as long as $k$ is smaller than a critical value $k_c = \rho J_0^2 / \left(8\sqrt{2\pi}a\right)$. These bump states are expressed as $\bar{U}(x) = A_U exp\left[-(x-z)^2 / \left(4a^2\right)\right]$, with $z$ a free parameter representing the bump center and $A_U$ a constant representing the bump amplitude.

In general, although the state of the network is affected by external inputs and adaptation, the network bump can still be well approximated with Gaussian-like profiles (*Mi et al., 2014*). Therefore, we assume the state of the network to be of the following Gaussian form:

$$\overline{U}(x,t) = A_u exp\left\{-\frac{[x-z(t)]^2}{4a^2}\right\}, \tag{A5}$$

$$\overline{V}(x,t) = A_v exp\left\{-\frac{[x-z(t)+d(t)]^2}{4a^2}\right\}, \tag{A6}$$

$$\bar{r}(x,t) = A_r exp\left\{-\frac{[x-z(t)]^2}{2a^2}\right\}, \tag{A7}$$

where $A_u$, $A_v$, and $A_r$ represent the amplitudes of these Gaussian bumps. $z(t)$ is the center of $U(x, t)$ and $r(x, t)$. $d(t)$ denotes the distance between $U(x, t)$ and $V(x, t)$, and $d(t) > 0$ always holds, as the firing rate adaptation is a much slower dynamics lagging behind the neural dynamics. Here, we assume that the bump heights, i.e., $A_u$, $A_v$, and $A_r$, are all constants during the evolution of the neural dynamics.

To solve the specific solution of the network state, we need to substitute the general solution given by *Equations A5–A7* into the network dynamics *Equations A1–A3*, we obtain:

$$A_r = \frac{A_u^2}{1 + k\rho\sqrt{2\pi}aA_u^2}, \tag{A8}$$

$$\tau\left[A_u\frac{x-z}{2a^2}\frac{dz}{dt} + \frac{dA_u}{dt}\right]\mathcal{N}(z, 2a) = \left(-A_u + \frac{\rho J_0}{\sqrt{2}}A_r\right)\mathcal{N}(z, 2a) \\ -A_v\mathcal{N}(z-d, 2a) + I^{ext}(x, t), \tag{A9}$$

$$\tau_v\left[A_v\frac{x-z+d}{2a^2}\frac{d(z-d)}{dt} + \frac{dA_v}{dt}\right]\mathcal{N}(z-d, 2a) = -A_v\mathcal{N}(z-d, 2a) \\ +mA_u\mathcal{N}(z, 2a), \tag{A10}$$

where $\mathcal{N}(z, 2a) = exp\left\{-[x-z]^2/4a^2\right\}$. Given $I^{ext}(x, t) = 0$, we can solve **Equations A8–A10** to get the solution of the network state. An important property of a CANN is that its dynamics is dominated by a few motion modes, as a consequence of the translation-invariant connections between neurons. We can therefore simplify the network dynamics significantly by projecting the network dynamics onto its dominating motion modes (**Fung et al., 2010**) (by projecting a function $f(x)$ onto a mode $u_n(x)$, it means to compute $\int_x f(x) u_n(x) dx$). Typically, projecting onto the first two motion modes is adequate (see next section).

For the bump $U(x, t)$, the first two motion modes are:

$$u_0(x, t) = exp\left\{-\frac{[x - z(t)]^2}{4a^2}\right\}, \tag{A11}$$

$$u_1(x, t) = [x - z(t)]\, exp\left\{-\frac{[x - z(t)]^2}{4a^2}\right\}. \tag{A12}$$

For the bump $V(x, t)$, the first two motion modes are:

$$v_0(x, t) = exp\left\{-\frac{[x - z(t) + d(t)]^2}{4a^2}\right\}, \tag{A13}$$

$$v_1(x, t) = [x - z(t) + d(t)]\, exp\left\{-\frac{[x - z(t) + d(t)]^2}{4a^2}\right\}. \tag{A14}$$

## 2 Deriving the network state when the external input does not exist ($I^{ext} = 0$)

### 2.1 Static bump state of the network

We first analyze the condition for the network holding a static bump as its stationary state. In this case, the positions of bumps $U$ and $V$ remain unchanged, i.e., $dz/dt = 0$, and the discrepancy between them is zero, i.e., $d = 0$. Thus, **Equations A9 and A10** can be simplified as:

$$\tau\frac{dA_u}{dt} = -A_u + \frac{\rho J_0}{\sqrt{2}}A_r - A_v, \tag{A15}$$

$$\tau_v\frac{dA_v}{dt} = -A_v + mA_u. \tag{A16}$$

Combining them with **Equation A8**, we obtain the solution of the steady state of the network as:

$$A_v = mA_u, \tag{A17}$$

$$A_r = \frac{\sqrt{2}(1 + m)}{\rho J_0}A_u, \tag{A18}$$

$$A_u = \frac{\rho J_0 + \sqrt{\rho^2 J_0^2 - 8\sqrt{2\pi}(1+m)^2 k\rho a}}{4\sqrt{\pi}(1+m)k\rho a}. \tag{A19}$$

To analyze the stability of this solution, we calculate the Jacobian matrix at this state, which is given by:

$$\mathbf{M} = \begin{pmatrix} \frac{1}{\tau}\left(-1 + \frac{\sqrt{2}\rho J_0 A_u}{\left(1 + \sqrt{2\pi}k\rho a A_u^2\right)^2}\right) & -\frac{1}{\tau} \\ \frac{m}{\tau_v} & -\frac{1}{\tau_v} \end{pmatrix} \tag{A20}$$

Denote the eigenvalues of the Jacobian matrix as $\lambda_1$ and $\lambda_2$. Therefore, the condition for the solution to be stable is that both eigenvalues are negative, which gives

$$\lambda_1 + \lambda_2 = \frac{1}{2}\left[-1 + \frac{\sqrt{2}A_u J_0 \rho}{\left(1 + \sqrt{2\pi}k\rho a A_u^2\right)^2} - \frac{\tau}{\tau_v}\right] < 0, \tag{A21}$$

$$\lambda_1 \lambda_2 = \frac{\tau}{\tau_v}\left(m + 1 - \frac{\sqrt{2}A_u J_0 \rho}{\left(1 + \sqrt{2\pi}k\rho a A_u^2\right)^2}\right) > 0. \tag{A22}$$

The above inequalities are satisfied when,

$$0 < k < k_{c1} = \frac{\rho J_0^2\left(1 + \frac{\tau}{\tau_v}\right)\left(1 + 2m - \frac{\tau}{\tau_v}\right)}{8\sqrt{2\pi}a\left(1 + m\right)^4}, \tag{A23}$$

$$0 < k < k_{c2} = \frac{\rho J_0^2}{8\sqrt{2\pi}a\left(1 + m\right)^2}. \tag{A24}$$

It is easy to check that $k_{c2} < k_{c1}$, so the condition for the network to hold static bumps as its steady state is $0 < k < k_{c2}$.

## 2.2 Traveling wave state of the network

We further analyze the condition for the network holding a continuously moving bump (traveling wave) as its steady state. In this state, the bump moves at a constant speed, and the center position is expressed as:

$$z(t) = v_{int}t, \tag{A25}$$

where $v_{int}$ is called the intrinsic speed of the bump activity. Since the bump height is roughly unchanged and the discrepancy $d$ is a constant, *Equations A9 and A10* can be simplified as:

$$\tau\left(A_u\frac{x-z}{2a^2}v_{int}\right)\mathcal{N}(z, 2a) = \left(-A_u + \frac{\rho J_0}{\sqrt{2}}A_r\right)\mathcal{N}(z, 2a) - A_v\mathcal{N}(z - d, 2a), \tag{A26}$$

$$\tau_v\left(A_v\frac{x-z+d}{2a^2}v_{int}\right)\mathcal{N}(z-d, 2a) = -A_v\mathcal{N}(z-d, 2a) + mA_u\mathcal{N}(z, 2a). \tag{A27}$$

In order to obtain the solution of variables of $A_u, A_v, A_r, d,$ and $v_{int}$, we reduce the dimensionality of the network dynamics by projecting *Equations A26 and A27* onto the dominant motion modes $u_0(x), u_1(x), v_0(x), v_1(x)$. First, projecting both sides of *Equation A26* onto the motion mode $u_0(x)$ (expressed in *Equation A11*), we obtain:

$$Left-side=0,$$

$$Right-side=\left(-A_u+\frac{\rho J_0}{\sqrt{2}}A_r\right)\sqrt{2\pi}a-A_v exp\left(-\frac{d^2}{8a^2}\right)\sqrt{2\pi}a.$$

Equating both sides, we have

$$-A_u+\frac{\rho J_0}{\sqrt{2}}A_r-A_v exp\left(-\frac{d^2}{8a^2}\right)=0. \tag{A28}$$

Similarly, projecting *Equation A26* onto the motion mode $u_1(x)$ (expressed in *Equation A12*) and equating both sides, we obtain:

$$\tau A_u v_{int}=dA_v exp\left(-\frac{d^2}{8a^2}\right). \tag{A29}$$

Again, projecting both sides of *Equation A27* onto the motion modes $u_0(x)$ and $u_1(x)$, respectively, and equating both sides, we obtain:

$$\frac{d}{4a^2}\tau_v A_v exp\left(-\frac{d^2}{8a^2}\right)v_{int}=-A_v exp\left(-\frac{d^2}{8a^2}\right)+mA_u, \tag{A30}$$

$$\tau_v\left(1-\frac{d^2}{4a^2}\right)v_{int}=d. \tag{A31}$$

Combining *Equation A8* and *Equations A28–A31*, we obtain the values of $A_u,A_v,A_r,d,$ and $v_{int}$ in the traveling wave state as follows:

$$A_u=\frac{\rho J_0+\sqrt{\rho^2 J_0^2-8\sqrt{2\pi}k\rho a\left(1+\sqrt{\frac{m\tau}{\tau_v}}\right)^2}}{4\sqrt{\pi}k\rho a\left(1+\sqrt{\frac{m\tau}{\tau_v}}\right)}, \tag{A32}$$

$$A_v=\frac{\rho J_0+\sqrt{\rho^2 J_0^2-8\sqrt{2\pi}k\rho a\left(1+\sqrt{\frac{m\tau}{\tau_v}}\right)^2}}{2\sqrt{2\pi}k\rho^2 a J_0}, \tag{A33}$$

$$A_r=\sqrt{\frac{m\tau}{\tau_v}}exp\left[\frac{1-\sqrt{\frac{\tau}{m\tau_v}}}{2}\right]\frac{\rho J_0+\sqrt{\rho^2 J_0^2-8\sqrt{2\pi}k\rho a\left(1+\sqrt{\frac{m\tau}{\tau_v}}\right)^2}}{4\sqrt{\pi}k\rho a\left(1+\sqrt{\frac{m\tau}{\tau_v}}\right)}, \tag{A34}$$

$$d=2a\sqrt{1-\sqrt{\frac{\tau}{m\tau_v}}}, \tag{A35}$$

$$v_{int}=\frac{2a}{\tau_v}\sqrt{\frac{m\tau_v}{\tau}-\sqrt{\frac{m\tau_v}{\tau}}}. \tag{A36}$$

It is straightforward to check from *Equation A36* that, to obtain a traveling wave state, $v_{int}$ should be a real positive value. This gives the condition that

$$m>\frac{\tau}{\tau_v}. \tag{A37}$$

*Equations A24 and A37* give the phase diagram of the network state when external input does not exist, as shown in *Appendix 1—figure 2g* in the main text.

## 3 Deriving the oscillatory tracking state of the network when the external input is applied ($I^{ext} \neq 0$)

The external input is given by:

$$I^{ext} = \alpha exp\left[-\frac{(x - v_{ext}t)^2}{4a^2}\right],\tag{A38}$$

where $\alpha$ is the input strength and $v_{ext}$ is the speed of the external input, mimicking the moving speed of the artificial animal.

The network state is mainly affected by two competing factors: one is the intrinsic mobility of the network originated from the firing rate adaptation, which drives the bump to move spontaneously (see section Traveling wave state of the network above); the other is the extrinsic mobility driven by the external input, which drives the bump to move at the same speed of $v_{ext}$. The competition between these two factors leads to three tracking states of the network: traveling wave, oscillatory tracking, and smooth tracking. Among the three states, the traveling wave state and the smooth tracking state are similar to the two cases we described in the previous section. Here, we only focus on the analytical derivation of the oscillatory tracking state. By numerically simulating the attractor network, we find that the bump center position can be roughly expressed as a sinusoidal moving wave given by:

$$z(t) = c_0 sin(\omega t) + d_0 + v_{ext}t,\tag{A39}$$

where $c_0$ and $\omega$ represent the amplitude and frequency of the sinusoidal wave, respectively, and $d_0$ denotes the offset between the center of the activity bump and the center of the external input. Substituting the form of external input expressed in *Equation A38* into *Equations A9 and A10*, we obtain:

$$\tau\left(A_u\frac{x - z}{2a^2}\frac{dz}{dt}\right)\mathcal{N}(z, 2a) = \left(-A_u + \frac{\rho J_0}{\sqrt{2}}A_r\right)\mathcal{N}(z, 2a) - A_v\mathcal{N}(z - d, 2a)$$
$$+\alpha\mathcal{N}(v_{ext}t, 2a),\tag{A40}$$

$$\tau_v\left(A_v\frac{x - z + d}{2a^2}\frac{d(z - d)}{dt}\right)\mathcal{N}(z - d, 2a) = -A_v\mathcal{N}(z - d, 2a) + mA_u\mathcal{N}(z, 2a).\tag{A41}$$

Again, to get the solution of variables of $A_u, A_v, A_r, c_0, \omega, d_0$ we reduce the dimensionality of the network dynamics by projecting *Equations A40 and A41* onto the dominant motion modes $u_0(x), u_1(x), v_0(x), v_1(x)$. Projecting *Equation A40* onto $u_0$ and $u_1$, respectively, gives

$$-A_u + \frac{\rho J_0}{\sqrt{2}}A_r + \alpha exp\left(-\frac{s^2}{8a^2}\right) = A_v exp\left(-\frac{d^2}{8a^2}\right),\tag{A42}$$

$$dA_v exp\left(-\frac{d^2}{8a^2}\right) - \alpha s exp\left(-\frac{s^2}{8a^2}\right) = \tau A_u\frac{dz}{dt},\tag{A43}$$

where $s(t) = c_0 sin(\omega t) + d_0$ denotes the offset between $U(x, t)$ and $I_{ext}(x, t)$. For clearance, we denote $A_{temp} = A_v exp\left(-d^2/8a^2\right)$ hereafter.

We first solve the dynamics of the distance $d(t)$, i.e., the distance between $U(x, t)$ and $V(x, t)$. To do this, we substitute *Equations A39 and A42* into *Equation A43* and obtain:

$$d(t) = \frac{1}{A_{temp}}\left[\tau A_u(v + c_0\omega cos\omega t) + \alpha s exp\left(-\frac{s^2}{8a^2}\right)\right].\tag{A44}$$

Since $s \ll 2a$ generally holds, we have $exp\left(-s^2/8a^2\right) \approx 1$. With this approximation, the equation above can be rewritten as:

$$d(t) = A_0 sin(\omega t + \beta) + B_0,\tag{A45}$$

where the parameters are solved as:

$$\beta = arccos\left(\frac{\alpha}{\sqrt{\tau^2 A_u^2 \omega^2 + \alpha^2}}\right), \tag{A46}$$

$$A_0 = \frac{c_0\sqrt{\tau^2 A_u^2 \omega^2 + \alpha^2}}{A_{temp}}, \tag{A47}$$

$$B_0 = \frac{\tau A_u v + \alpha d_0}{A_{temp}}. \tag{A48}$$

Again, projecting *Equation A41* onto $v_0$ and $v_1$, respectively, gives

$$A_v = mA_u exp\left(-\frac{d^2}{8a^2}\right), \tag{A49}$$

$$\tau_v A_v \left[\frac{dz(t)}{dt} - \frac{dd(t)}{dt}\right] = mA_u exp\left(-\frac{d^2}{8a^2}\right) d(t). \tag{A50}$$

To solve the expression of $A_u$, we substitute *Equations A8 and A49* into *Equations A42* and obtain:

$$-A_u + \frac{\rho J_0}{\sqrt{2}} \frac{A_u^2}{1 + \sqrt{2\pi}ak\rho A_u^2} + \alpha exp\left(-\frac{s^2}{8a^2}\right) = mA_u exp\left(-\frac{d^2}{4a^2}\right). \tag{A51}$$

Since $s \ll 2a$ and $d \ll 2a$, the approximations of $exp\left(-s^2/8a^2\right) \approx 1$ and $exp\left(-d^2/4a^2\right) \approx 1$ hold, and the above equation can be simplified as:

$$(m+1)A_u - \frac{\rho J_0}{\sqrt{2}} \frac{A_u^2}{1 + \sqrt{2\pi}ak\rho A_u^2} - \alpha = 0. \tag{A52}$$

We can rearrange *Equation A52* into a general cubic equation of $A_u$, which is written as:

$$a_3 A_u^3 + a_2 A_u^2 + a_1 A_u + a_0 = 0, \tag{A53}$$

$$a_3 = \sqrt{2\pi}(m+1)ak\rho, \tag{A54}$$

$$a_2 = -\sqrt{2\pi}ak\rho\alpha - \frac{\rho J_0}{\sqrt{2}}, \tag{A55}$$

$$a_1 = m+1, \tag{A56}$$

$$a_0 = -\alpha, \tag{A57}$$

It's easy to check that *Equation A53* only has one real solution, which is:

$$A_u = \left[-\frac{q}{2} + \sqrt{\left(\frac{q}{2}\right)^2 + \left(\frac{p}{3}\right)^3}\right]^{1/3} + \left[-\frac{q}{2} - \sqrt{\left(\frac{q}{2}\right)^2 + \left(\frac{p}{3}\right)^3}\right]^{1/3}, \tag{A58}$$

$$q = \frac{3a_3 a_1 - a_2^2}{3a_3^2}, \tag{A59}$$

$$p = \frac{27a_3^2 a_0 - 9a_3 a_2 a_1 + 2a_2^3}{27a_3^3}. \tag{A60}$$

The analytical solution of $A_u$ given by *Equations A58–A60* is very complicated. However, by numerically simulating the network, we find that $\sqrt{2\pi}ak\rho A_u^2 \gg 1$ can be hold when the network is at the oscillatory tracking state. This gives $A_u^2/\left(1 + \sqrt{2\pi}ak\rho A_u^2\right) \approx 1/\sqrt{2\pi}ak\rho$. Therefore, we can simplify the expression of $A_u$ to

$$A_u = \frac{J_0 + 2\sqrt{\pi}ak\alpha}{2\sqrt{\pi}ak\left(1 + m\right)}. \tag{A61}$$

To solve $\omega, d_0$, and $c_0$, we further substitute *Equation A49* into *Equation A50* and obtain:

$$\tau_v \frac{\mathrm{d}z\left(t\right)}{\mathrm{d}t} = d\left(t\right) + \tau_v \frac{\mathrm{d}d\left(t\right)}{\mathrm{d}t}. \tag{A62}$$

The above equation can be expanded by substituting *Equations A39 and A45* into *Equation A62* which gives

$$\tau_v\left(v + c_0\omega cos\omega t\right) = A_0\left[sin\left(\omega t + \beta\right) + \omega\tau_v cos\left(\omega t + \beta\right)\right] + B_0.$$

Using the trigonometric transformation formula, we can rewrite the above equation as:

$$\tau_v v + \tau_v c_0\omega sin\left(\omega t + \frac{\pi}{2}\right) = A_0\sqrt{1 + \omega^2\tau_v^2}sin\left(\omega t + \beta + \gamma\right) + B_0, \tag{A63}$$

where $\gamma$ is given by:

$$\gamma = arccos\left(\frac{1}{\sqrt{\tau_v^2\omega^2}}\right). \tag{A64}$$

Equating two sides of *Equation A63*, we have:

$$\tau_v v = B_0, \tag{A65}$$

$$\frac{\pi}{2} = \beta + \gamma, \tag{A66}$$

$$\tau_v c_0\omega = A_0\sqrt{1 + \omega^2\tau_v^2}. \tag{A67}$$

Now we combine the three equations given by *Equations A65–A67* to get the solutions of $c_0, \omega$, and $d_0$. Substituting *Equation A48* into *Equation A65*, we obtain:

$$d_0 = \frac{\tau_v v A_{temp} - \tau v A_u}{\alpha}. \tag{A68}$$

Applying the cosine function to both sides of *Equation A66*, we obtain:

$$cos\left(\beta + \gamma\right) = cos\gamma cos\beta - sin\gamma sin\beta = 0. \tag{A69}$$

Substituting *Equations A46 and A64* into *Equation A69*, we have:

$$\omega^2 = \frac{\alpha}{\tau\tau_v A_u}. \tag{A70}$$

Combining *Equation A70* with *Equation A61*, we obtain the expression for the oscillating frequency $\omega$, i.e.,

$$\omega = \sqrt{\frac{2\sqrt{\pi}\alpha ak\left(1 + m\right)}{\tau\tau_v\left(J_0 + 2\sqrt{\pi}ak\alpha\right)}}. \tag{A71}$$

Substituting *Equations A47 and A70* into *Equation A67*, and taking square on both sides, we have:

$$\frac{\left(\tau^2 A_u^2 \omega^2 + \alpha^2\right)}{A_{temp}^2}\left(1 + \omega^2 \tau_v^2\right) = \tau_v^2 \omega^2. \tag{A72}$$

Solving the above equation for $A_{temp}$, we get:

$$A_{temp} = \frac{\tau A_u + \alpha \tau_v}{\tau_v}. \tag{A73}$$

Substituting *Equation A73* into *Equation A68*, we can get the expression for the average offset $d_0$ between the bump center and the external input center, which is:

$$d_0 = \tau_v v. \tag{A74}$$

Since $A_{temp} = m A_u exp\left[-d\left(t\right)^2 / \left(4a^2\right)\right]$ varies across time, we take the approximation

$$A_{temp} = m A_u exp\left[-\overline{d\left(t\right)^2}/(4a^2)\right], \tag{A75}$$

with $\overline{d\left(t\right)^2}$ the time-averaged value, which is calculated to be:

$$\overline{d\left(t\right)^2} = \frac{1}{T}\int_0^T d^2\left(t\right)dt = \frac{\alpha \tau_v}{2\left(\tau A_u + \alpha \tau_v\right)}c_0^2 + \tau_v^2 v^2. \tag{A76}$$

Substituting *Equations A73 and A76* into *Equation A75*, we obtain the expression for the oscillating amplitude $c_0$, i.e.,

$$c_0 = \sqrt{\frac{2\left(\tau A_u + \alpha \tau_v\right)}{\alpha \tau_v}\left[4a^2\left(ln\frac{\tau_v m A_u}{\tau A_u + \alpha \tau_v}\right) - \tau_v^2 v^2\right]}, \tag{A77}$$

Substituting *Equation A73* into *Equation A49* and utilizing the condition of $exp\left(-d^2/8a^2\right) = \sqrt{A_{temp}/m A_u}$, we obtain the expression for the bump height of $V\left(x, t\right)$:

$$A_v = \sqrt{\left(\frac{\tau A_u + \alpha \tau_v}{\tau_v}\right)m A_u}. \tag{A78}$$

Overall, combining *Equations A8, A61, A71, A74, A77, and A78*, we get the solutions of all variables in the oscillatory tracking state, which are expressed as:

$$A_u = \frac{J_0 + 2\sqrt{\pi}ak\alpha}{2\sqrt{\pi}ak\left(1 + m\right)}, \tag{A79}$$

$$A_r = \frac{A_u^2}{1 + \sqrt{2\pi}ak\rho A_u^2}, \tag{A80}$$

$$A_v = \sqrt{\left(\frac{\tau A_u + \alpha \tau_v}{\tau_v}\right)m A_u}, \tag{A81}$$

$$c_0 = A_v\sqrt{\frac{2}{\alpha m A_u}\left[8a^2 ln\frac{m A_u}{A_v} - \tau_v^2 v^2\right]}, \tag{A82}$$

$$d_0 = \tau_v v, \tag{A83}$$

$$\omega = \sqrt{\frac{\alpha}{\tau \tau_v A_u}}. \tag{A84}$$

It is noteworthy that the theoretical solutions given by *Equations A82–A84* exist only if the sweep amplitude $c_0$ given by *Equation A82* is of real value. This gives the condition for the CANN to be in the oscillatory tracking state, which is:

$$8a^2 ln \frac{mA_u}{A_v} > \tau_v^2 v^2. \tag{A85}$$

We carry out numerical simulations to verify our theoretical results, including the theoretical solutions of the mean offset $d_0$ given by *Equation A83* (see *Appendix 1—figure 2a–c*) and the theoretical boundary that separates the smooth tracking state and the oscillatory tracking state given by *Equation A85* (see *Appendix 1—figure 2d*). The results show that our theoretical analysis agrees well with the simulation results.

## 4 Oscillatory tracking in the 2D CANN – modeling theta sweeps in the open field environment

### 4.1 Model description

In the 2D CANN, neurons are uniformly distributed on a rectangular neuronal sheet arranged according to the locations of their firing fields. Denote $U(\mathbf{x}, t)$ as the synaptic input to the neuron at location $\mathbf{x}$, with $\mathbf{x} = (x_1, x_2)$ and $x_1, x_2 \in (-\infty, \infty)$, and $r(\mathbf{x}, t)$ as the corresponding firing rate. The dynamics of $U(\mathbf{x}, t)$ is determined by its own relaxation, the recurrent inputs from other neurons, and the firing rate adaptation, which is written as:

$$\tau \frac{\partial U(\mathbf{x}, t)}{\partial t} = -U(\mathbf{x}, t) + \rho \int_{\mathbf{x}'} J(\mathbf{x}, \mathbf{x}') r(\mathbf{x}', t) \, d\mathbf{x}' - V(\mathbf{x}, t) + \sigma_U \xi_U(\mathbf{x}, t). \tag{A86}$$

Here, $\tau$ is the time constant of synaptic current and $\rho$ is the neuronal density. The recurrent connection is defined as: $J(\mathbf{x}, \mathbf{x}') = J_0 / (2\pi a^2) exp\left[-\|\mathbf{x} - \mathbf{x}'\|^2 / (2a^2)\right]$, with $\|\mathbf{x} - \mathbf{x}'\|^2 = (x_1 - x_1')^2 + (x_2 - x_2')^2$ which is translation-invariant on the neuronal sheet (*Appendix 1—figure 6a*). The nonlinear relationship between the firing rate $r(\mathbf{x}, t)$ and the synaptic input $U(\mathbf{x}, t)$ is implemented by the divisive normalization, which is:

$$r(\mathbf{x}, t) = \frac{U^2(\mathbf{x}, t)}{1 + k\rho \int_{\mathbf{x}'} U^2(\mathbf{x}', t) \, d\mathbf{x}'}, \tag{A87}$$

where $k$ controls the normalization strength. In the neural system, divisive normalization could be implemented by shunting inhibition (*Mitchell and Silver, 2003*). The term $V(\mathbf{x}, t)$ on the right-hand side of *Equation A86* represents the effect of firing rate adaptation, with the dynamics written as:

$$\tau_v \frac{\partial V(\mathbf{x}, t)}{\partial t} = -V(\mathbf{x}, t) + mU(\mathbf{x}, t), \tag{A88}$$

where $\tau_v$ is the time constant, and $m$ is the adaptation strength.

### 4.2 Theta sweeps in the 2D CANN

We study the network dynamics when an moving external input is applied to the network, which is written as:

$$I^{ext} = \alpha exp\left[-\frac{\|\mathbf{x} - \mathbf{v_{ext}} t\|^2}{4a^2}\right], \tag{A89}$$

where $\mathbf{v_{ext}} = (v_x, v_y)$ represents the speed of the external input and $\alpha$ represents the input strength. We consider one simple case, where the external input is moving on a straight line along the x-axis, i.e., $v_y = 0$. We find that, similar to 1D CANN, when the input strength $\alpha$ and adaptation strength $m$ is set appropriately, the bump activity also oscillates around the moving input along the moving direction, which give rise to the alternative forward and reverse theta sequences along moving direction of the external input (*Appendix 1—figure 6b*). The heatmap of the phase shift of the probe neuron which is located at $x = 0, y = 0$ is shown in *Appendix 1—figure 6c*.

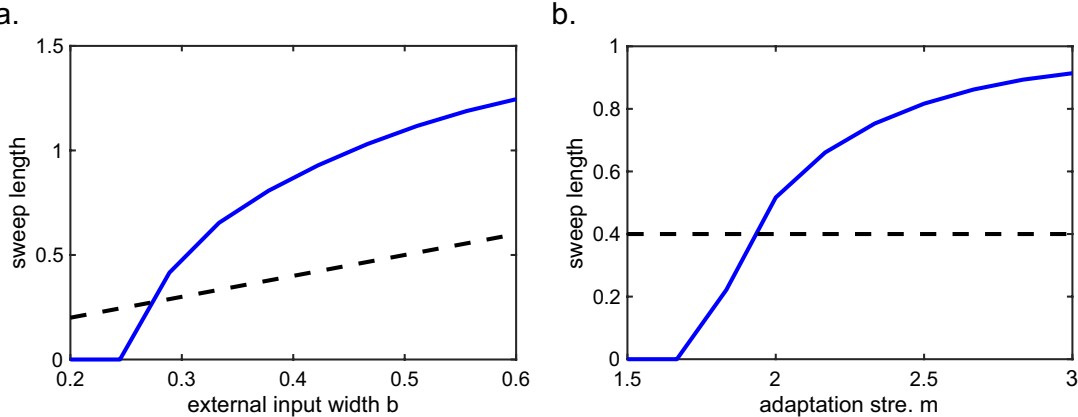

**Appendix 1—figure 1.** Sweep length is not bounded by the external input width. (**a**) The sweep length is positively but not linearly related with the external input width. (**b**) With fixed external input width, increasing the adaptation strength the sweep length can exceed the external input width. This figure relates to *Figure 2* in the main text.

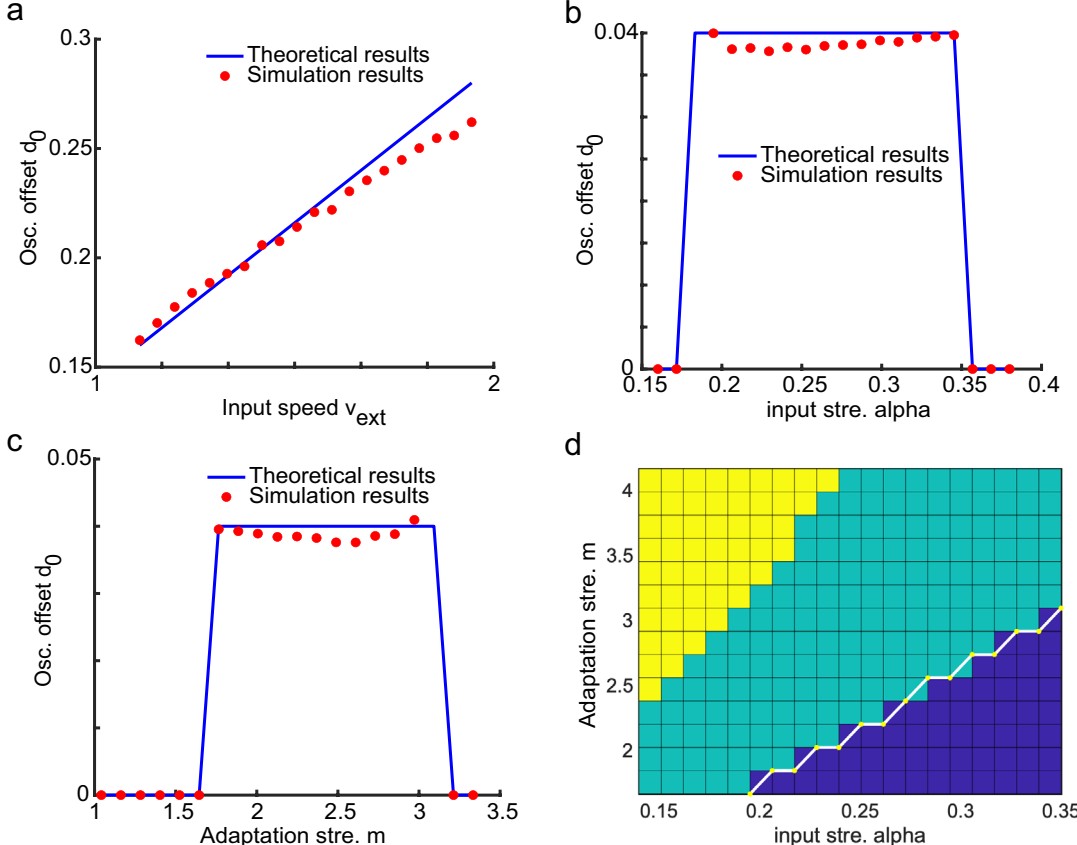

**Appendix 1—figure 2.** Verifying theoretical results with numerical simulations. (**a–c**) Simulation results of the average offset $d_0$ as a function of $v_{ext}$, $\alpha$, and $m$, respectively. (**d**) The phase diagram of network states. The yellow area represents the traveling wave state, the green area represents the oscillatory tracking state, and the blue area represents the smooth tracking state. The white line represents the theoretical boundary given by *Equation A85*. The parameters used in simulations are: $k = 5$, $J_0 = 1$, $a = 0.4$, $N = 512$, $\tau = 3$ ms, $\tau_v = 144$ ms, $\rho = 20.37$. This figure relates to *Figure 2* in the main text.

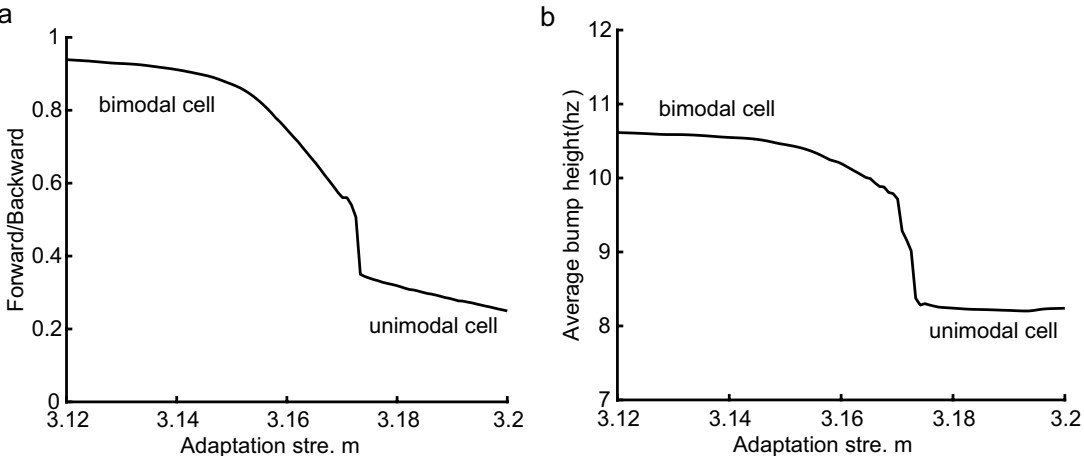

**Appendix 1—figure 3.** Activity bump height as a function of the adaptation strength. (**a**) The ratio between the average bump height during forward window and the average bump height during backward window as function of the adaptation $m$. When the adaptation strength is relatively small, the mean firing rate of place cells is approximately the same in the forward window as in the backward window. And the place cells exhibit bimodal cell properties. As the adaptation strength gets larger, the mean firing rates in the backward window gradually decrease and the place cells tend to exhibit firing properties more like unimodal cells. (**b**) The average bump height as a function of the adaptation strength $m$. Our model predicts that the bimodal cells fire at higher frequency than unimodal cells which can be testable in future experiments. The parameters are: $\alpha = 0.19$, $k = 5$, $J_0 = 1$, $a = 0.4$, $N = 512$, $\tau = 3\,\mathrm{ms}$, $\tau_v = 144\,\mathrm{ms}$, $\rho = 20.37$, $v_{ext} = 0.51\,\mathrm{m/s}$. This figure relates to **Figure 4** in the main text.

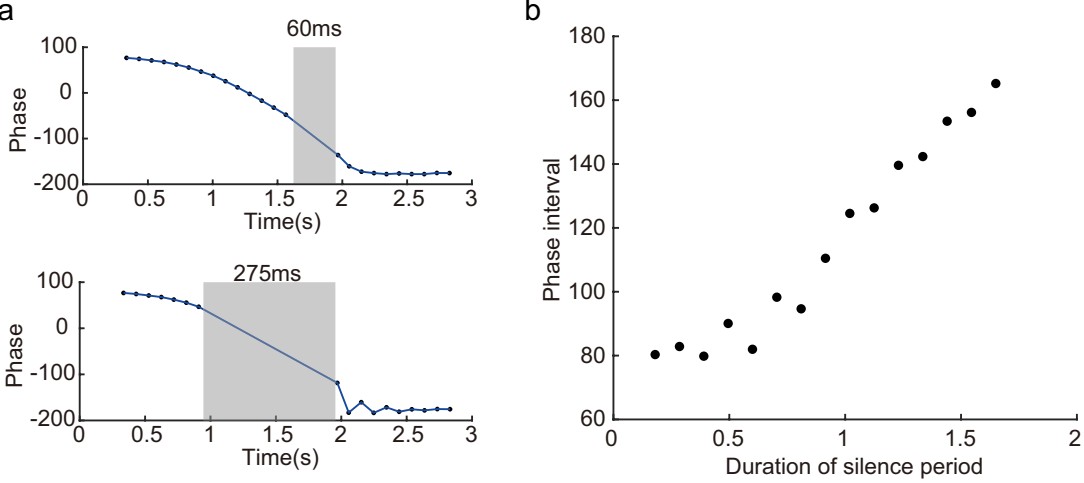

**Appendix 1—figure 4.** Persistent phase shift with variable silencing periods. (**a**) Two examples of the persisting phase shift after transient silencing. Upper panel: The silencing duration is 60 ms. Upper panel: The silencing duration is 275 ms. (**b**) The phase interval before and after the silencing as a function of the duration of the silencing. The phase interval gradually increases with the silencing duration. The parameters are: $\alpha = 0.19$, $m = 3.23$, $a = 0.4$, $k = 5$, $J_0 = 1$, $N = 512$, $\tau = 3\,\mathrm{ms}$, $\tau_v = 144\,\mathrm{ms}$, $\rho = 20.37$, $v_{ext} = 1\,\mathrm{m/s}$. This figure relates to **Figure 6** in the main text.

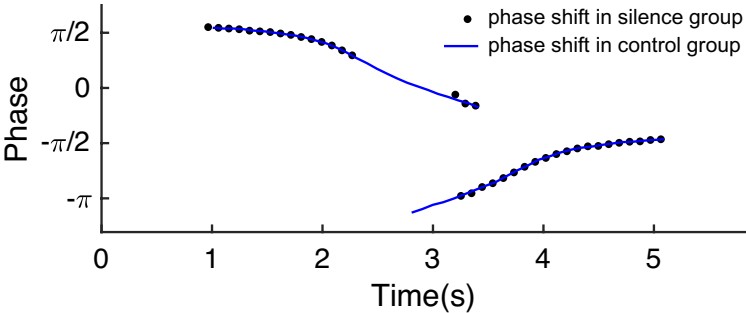

**Appendix 1—figure 5.** Persistent bimodal phase shift after transient silencing. The parameters are: $\alpha = 0.19$, $m = 3.03$, $a = 0.4$, $k = 5$, $J_0 = 1$, $N = 512$, $\tau = 3\,\text{ms}$, $\tau_v = 144\,\text{ms}$, $\rho = 20.37$, $v_{ext} = 0.3\,\text{m/s}$. This figure relates to **Figure 6** in the main text.

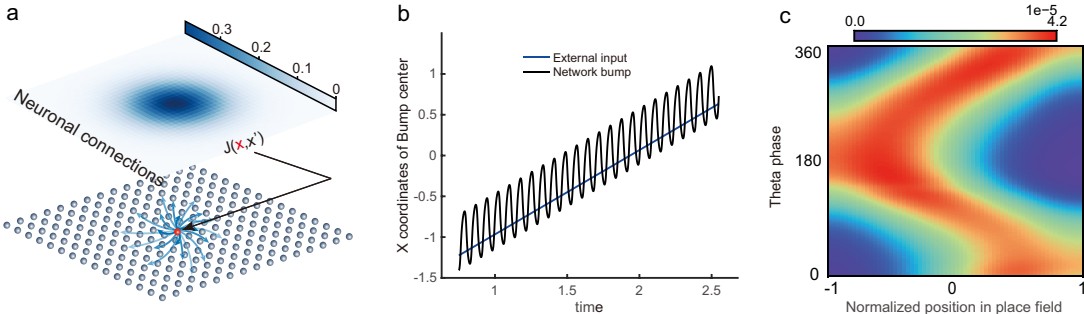

**Appendix 1—figure 6.** Theta sweeps and theta phase shift in a two-dimensional (2D) continuous attractor neural network (CANN). (**a**) A demonstration of the 2D CANN. (**b**) The trajectory of the bump center and external input center when the input is moving along the x-axis in the 2D CANN. (**c**) Theta phase as a function of the normalized position of the animal in place field, averaged over all place cells that are placed on the x-axis. –1 represents that the animal just enters the place field, and 1 represents that the animal is about to leave the place field. The parameters are: $\alpha = 0.2$, $m = 4$, $k = 6$, $J_0 = 1$, $a = 0.3$, $N = 16384$, $\tau = 3\,\text{ms}$, $\tau_v = 150\,\text{ms}$, $\rho = 415.01$, $v_{ext} = 1\,m/s$. This figure relates to **Figure 2** in the main text.

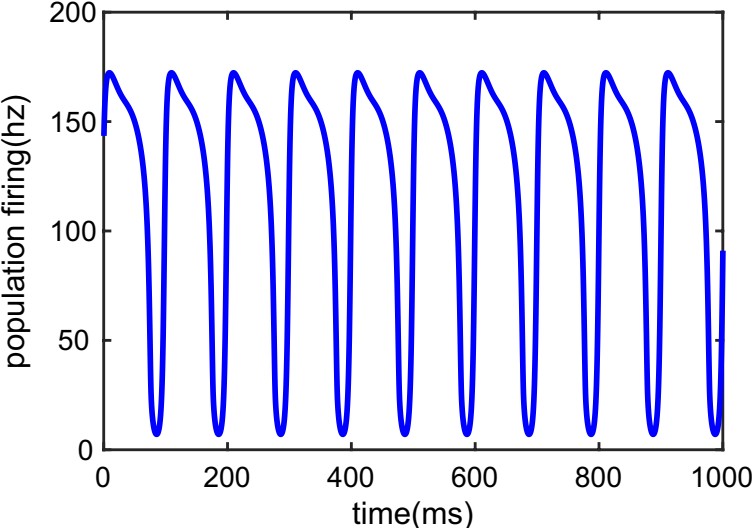

**Appendix 1—figure 7.** Theta oscillation of the population activities during the theta sweep state. This figure relates to **Figure 4** in the main text.

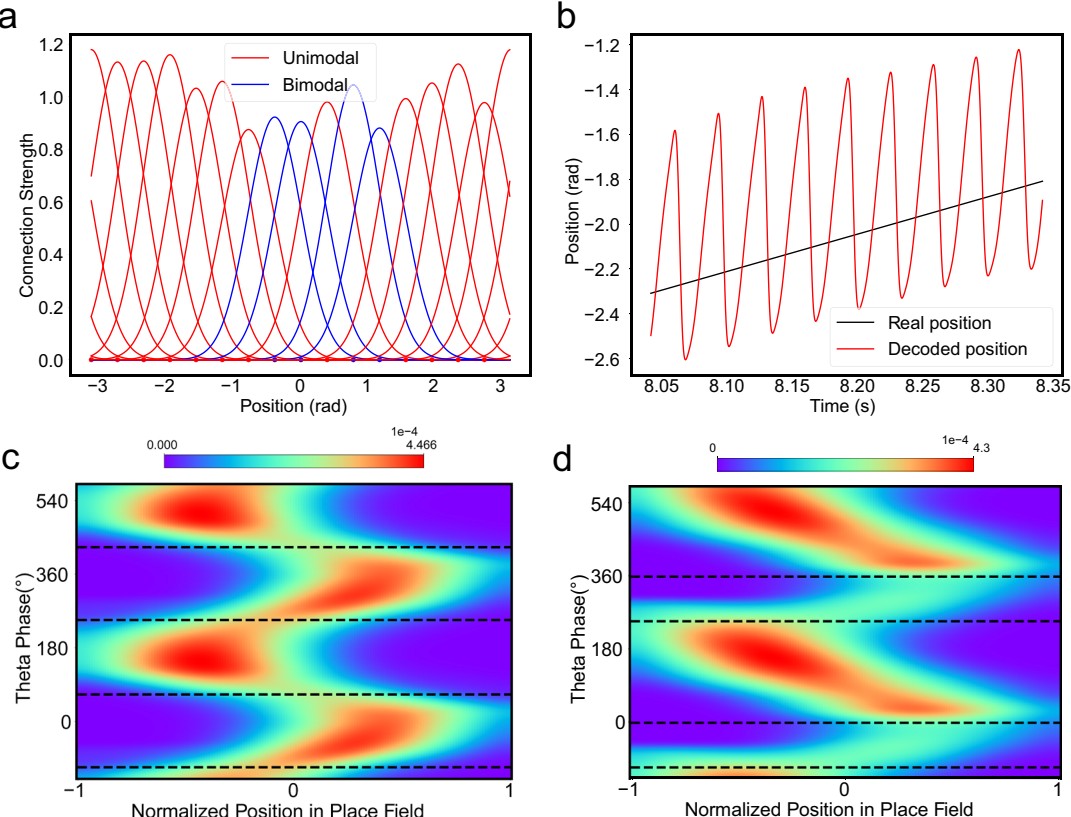

**Appendix 1—figure 8.** A-continuous attractor neural network (CANN) with heterogeneous connection strength generate oscillatory tracking to account for theta phase shift. (**a**) The synaptic connection strength profile of the neurons in the network. The blue lines represent the synaptic strengths of the neurons which turn out to be bimodal neurons, while the red lines represent the unimodal neurons. (**b**) The oscillatory tracking trajectory of the bump center. (**c**) The phase shift distribution of the bimodal cells. (**d**) The phase shift distribution of the unimodal cells. The variations of the connection strength is 0.1, the average value is 1. Other parameters are the same with *Figure 3* and *Figure 4* in the main text. This figure relates to *Figure 4* in the main text.

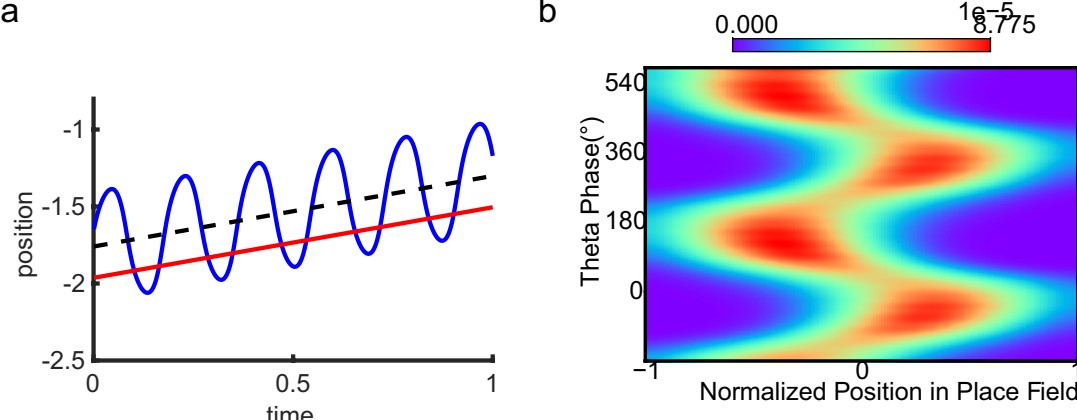

**Appendix 1—figure 9.** Oscillatory tracking behavior accounts for theta phase shift with $\tau = 10\,\mathrm{ms}$. (**a**) Oscillatory tracking behavior. (**b**) Bimodal phase shift of one example neuron. $\tau = 10\,\mathrm{ms}$, $\tau_v = 480\,\mathrm{ms}$, $\alpha = 0.3$, $m = 6$, Other parameters are set equal with the *Figure 2* in the main text. This figure relates to *Figure 2* in the main text.

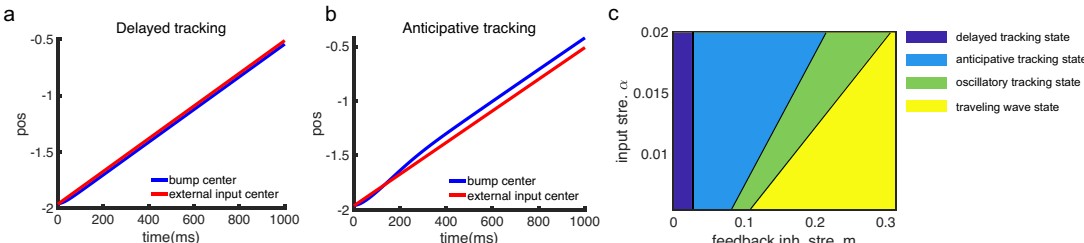

**Appendix 1—figure 10.** Spatiotemporal tracking dynamics when the adaptation strength is low (start from 0). (**a**) The tracking behavior when the adaptation strength $m = 0$ ($\alpha = 0.02$). The network bump can generate a bump to track the external input stimuli smoothly but with a constant lagging distance which is proportional to the time constants $\tau$ and the external input speed $v_{ext}$. (**b**) The tracking behavior when the adaptation strength $m = 0.1$ ($\alpha = 0.02$). Thanks to the intrinsic mobility introduced by SFA, the network bump anticipatively track the external input stimuli with a contant leading distance which is proportional to the adaptation strength $m$ and the external input speed $v_{ext}$. (**c**) A phase diagram that summarizes the spatiotemporal patterns of the A-CANN tracking behavior. Other parameters are set equal with **Figure 2** in the main text. This figure relates to **Figure 2** in the main text.

**Appendix 1—table 1.** Commonly used parameter values in the simulation of the linear track environment.

| Parameters | Values |
| --- | --- |
| Number of place cells: $N$ | 512 |
| Time constant of neural firing: $\tau$ | 3 ms |
| Time constant of spike frequency adaptation: $\tau_v$ | 144 ms |
| Neuron density: $\rho$ | $256/\pi$ |
| Recurrent connection range (Gaussian width): $a$ | 0.4 m |
| Width of external input (Gaussian width): $\sigma$ | 0.4 m |
| Recurrent connection strength: $J_0$ | 0.2 |
| Gain factor: $g$ | 5 |
| Global inhibition strength: $k$ | 5 |
| Moving speed of the external input: $v_{ext}$ (m/s) | 1.5 |
| Time interval: $\delta t$ | 0.3 s |
| Simulation duration: $T$ | 10 s |

**Appendix 1—table 2.** Figure-specific parameter values for input strength $\alpha$ and adaptation strength $m$.

| Figures/parameters | $\alpha$ | $m$ |
| --- | --- | --- |
| An example of smooth tracking (**Appendix 1—figure 2c**) | 0.19 | 0 |
| An example of traveling wave (**Appendix 1—figure 2d**) | 0 | 0.31 |
| Intrinsic speed vs. adaptation strength (**Appendix 1—figure 2e**) | 0 | 0:0.05:0.1 |
| Phase diagram (**Appendix 1—figure 2g**) | 0.05:0.001:0.16 | 0.9:0.01:1.8 |
| Oscillatory tracking (bimodal) (**Appendix 1—figure 4a, e, g**) | 0.19 | 3.02 |
| Oscillatory tracking (unimodal) (**Appendix 1—figure 4b, f, h**) | 0.19 | 3.125 |

*Appendix 1—table 3 Continued on next page*

**Appendix 1—table 3.** Parameters values in the simulation of the T-maze environment.

| Parameters | Values |
| --- | --- |
| Number of cells central/left/right: $N_1, N_2, N_3$ | 3000/1500/1500 |
| Time constant of neural firing: $\tau$ | 3 ms |
| Time constant of spike frequency adaptation: $\tau_v$ | 144 ms |
| Neuron density: $\rho$ | $(128/\pi)^2$ |
| Recurrent connection range (Gaussian width): $a$ | 0.3 |
| Recurrent connection strength: $J_0$ | $1.25 * 10^{-2}$ |
| Gain factor: $g$ | 20 |
| Global inhibition strength: $k$ | 1.25 |
| Moving speed of the external input: $v_{ext}$ (m/s) | 1.5 |
| Input strength: $\alpha$ | 2 |
| Adaptation strength: $m$ | 3.96 |
| Time interval: $\delta t$ | 0.3 s |
| Simulation duration: $T$ | 3.3 s |

