## [Editor Report · eLife assessment]

This study provides **valuable** new insights on how a prevailing model of hippocampal sequence formation can account for recent data, including forward and backward sweeps, as well as constant cycling of sweeps across different arms of a T-maze. The **convincing** evidence presented in support of this work relies on classical analytical and computational techniques about continuous attractor networks.

---

## [Referee Report · Reviewer #1 (Public review)]

Continuous attractor networks endowed with some sort of adaptation in the dynamics, whether that be through synaptic depression or firing rate adaptation, are fast becoming the leading candidate models to explain many aspects of hippocampal place cell dynamics, from hippocampal replay during immobility to theta sequences during run. Here, the authors show that a continuous attractor network endowed with spike frequency adaptation and subject to feedforward external inputs is able to account for several previously unaccounted aspects of theta sequences, including (1) sequences that move both forwards and backwards, (2) sequences that alternate between two arms of a T-maze, (3) speed modulation of place cell firing frequency, and (4) the persistence of phase information across hippocampal inactivations.

I think the main result of the paper (findings (1) and (2)) are likely to be of interest to the hippocampal community, as well as to the wider community interested in mechanisms of neural sequences. In addition, the manuscript is generally well written and the analytics are impressive. However, several issues should be addressed, which I outline below.

Major comments:

In real data, population firing rate is strongly modulated by theta (i.e., cells collectively prefer a certain phase of theta - see review paper Buzsaki, 2002) and largely oscillates at theta frequency during run. With respect to this cyclical firing rate, theta sweeps resemble "Nike" check marks, with the sweep backwards preceding the sweep forwards within each cycle before the activity is quenched at the end of the cycle. I am concerned that (1) the summed population firing rate of the model does not oscillate at theta frequency, and (2) as the authors state, the oscillatory tracking state must begin with a forward sweep. With regards to (1), can the authors show theta phase spike preference plots for the population to see if they match data? With regards to (2), can the authors show what happens if the bump is made to sweep backwards first, as it appears to do within each cycle?

I could not find the width of the external input mentioned anywhere in the text or in the table of parameters. The implication is that it is unclear to me whether, during the oscillatory tracking state, the external input is large compared to the size of the bump, so that the bump lives within a window circumscribed by the external input and so bounces off the interior walls of the input during the oscillatory tracking phase, or whether the bump is continuously pulled back and forth by the external input, in which case it could be comparable to the size of the bump. My guess based on Fig 2c is that it is the latter. Please clarify and comment.

I would argue that the "constant cycling" of theta sweeps down the arms of a T-maze was roughly predicted by Romani & Tsodyks, 2015, Figure 7. While their cycling spans several theta cycles, it nonetheless alternates by a similar mechanism, in that adaptation (in this case synaptic depression) prevents the subsequent sweep of activity from taking the same arm as the previous sweep. I believe the authors should cite this model in this context and consider the fact that both synaptic depression and spike frequency adaptation are both possible mechanisms for this phenomenon. But I certainly give the authors credit for showing how this constant cycling can occur across individual theta cycles.

The authors make an unsubstantiated claim in the paragraph beginning with line 413 that the Tsodyks and Romani (2015) model could not account for forwards and backwards sweeps. Both the firing rate adaptation and synaptic depression are symmetry breaking models that should in theory be able to push sweeps of activity in both directions, so it is far from obvious to me that both forward and backward sweeps are not possible in the Tsodyks and Romani model. The authors should either prove that this is the case (with theory or simulation) or excise this statement from the manuscript.

The section on the speed dependence of theta (starting with line 327) was very hard to understand. Can the authors show a more graphical explanation of the phenomenon? Perhaps a version of Fig 2f for slow and fast speeds, and point out that cells in the latter case fire with higher frequency than in the former?

I had a hard time understanding how the Zugaro et al., (2005) hippocampal inactivation experiment was accounted for by the model. My intuition is that while the bump position is determined partially by the location of the external input, it is also determined by the immediate history of the bump dynamics as computed via the local dynamics within the hippocampus (recurrent dynamics and spike rate adaptation). So that if the hippocampus is inactivated for an arbitrary length of time, there is nothing to keep track of where the bump should be when the activity comes back on line. Can the authors please explain more how the model accounts for this?

Can the authors comment on why the sweep lengths oscillate in the bottom panel of Fig 5b during starting at time 0.5 seconds before crossing the choice point of the T-maze? Is this oscillation in sweep length another prediction of the model? If so, it should definitely be remarked upon and included in the discussion section.

Perhaps I missed this, but I'm curious whether the authors have considered what factors might modulate the adaptation strength. In particular, might rat speed modulate adaptation strength? If so, would have interesting predictions for theta sequences at low vs high speeds.

I think the paper has a number of predictions that would be especially interesting to experimentalists but are sort of scattered throughout the manuscript. It would be beneficial to have them listed more prominently in a separate section in the discussion. This should include (1) a prediction that the bump height in the forward direction should be higher than in the backward direction, (2) predictions about bimodal and unimodal cells starting with line 366, (3) prediction of another possible kind of theta cycling, this time in the form of sweep length (see comment above), etc.

---

## [Referee Report · Reviewer #2 (Public review)]

In this work, the authors elaborate on an analytically tractable, continuous-attractor model to study an idealized neural network with realistic spiking phase precession/procession. The key ingredient of this analysis is the inclusion of a mechanism for slow firing-rate adaptation in addition to the otherwise fast continuous-attractor dynamics. The latter continuous-attractor dynamics classically arises from a combination of translation invariance and nonlinear rate normalization.

For strong adaptation/weak external input, the network naturally exhibits an internally generated, travelling-wave dynamics along the attractor with some characteristic speed. For small adaptation/strong external stimulus, the network recovers the classical externally driven continuous-attractor dynamics. Crucially, when both adaptation and external input are moderate, there is a competition with the internally generated and externally generated mechanisms leading to an oscillatory tracking regime. In this tracking regime, the population firing profile oscillates around the neural field tracking the position of the stimulus. The authors demonstrate by a combination of analytical and computational arguments that oscillatory tracking corresponds to realistic phase precession/procession. In particular the authors can account for the emergence of unimodal and bimodal cells, as well as some other experimental observations with respect the dependence of phase precession/procession on the animal's locomotion.

The strengths of this work are at least three-fold: (1) Given its simplicity, the proposed model has a surprisingly large explanatory power of the various experimental observations. (2) The mechanism responsible for the emergence of precession/procession can be understood as a simple yet rather illuminating competition between internally driven and externally driven dynamical trends. (3) Amazingly, and under some adequate simplifying assumptions, a great deal of analysis can be treated exactly, which allows for a detailed understanding of all parametric dependencies. This exact treatment culminates with a full characterization of the phase space of the network dynamics, as well as the computation of various quantities of interest, including characteristic speeds and oscillating frequencies.

As mentioned by the authors themselves, the main limitation of this work is that it deals with a very idealized model and it remains to see how the proposed dynamical behaviors would persists in more realistic models. For example, the model is based on a continuous attractor model that assumes perfect translation-invariance of the network connectivity pattern. Would the oscillating tracking behavior persist in the presence of connection heterogeneities? Another limitation is that the system needs to be tuned to exhibit oscillation within the theta range and that this tuning involves a priori variable parameters such as the external input strength. Is the oscillating-tracking behavior overtly sensitive to input strength variations? The author mentioned that an external pacemaker can serve to drive oscillation within the desired theta band but there is no evidence presented supporting this. A final and perhaps secondary limitation has to do with the choice of parameter, namely the time constant of neural firing which is chosen around 3ms. This seems rather short given that the fast time scale of rate models (excluding synaptic processes) is usually given by the membrane time constant, which is typically about 15ms. I suspect this latter point can easily be addressed.

---

## [Author Response]

The following is the authors’ response to the previous reviews.

**Response to reviewer #1:**

We thank the reviewer for the further recommendations for improving our presentation. We would like to carefully address the remaining concerns of the reviewer.

(1) I realize now that I didn't make my point clear enough, which was that as far as I know there is no reason to believe that an oscillatory state cannot be induced with synaptic depression as with spike frequency adaptation when used in the context of the author's model. I'm fine with how the authors have distinguished their model from R&T 2015, but I think the more interesting question is whether there is any reason to believe that STD is not equally capable of doing all the things mentioned in this paper as SFA, and if not why not. I would like the authors to go out on a limb and address this, if only with a few sentences in the discussion.

Thank you for pointing this out again. In response to your query regarding the comparison between STD and SFA in generating bump sweeps, we have done simulations based on STD. The results showed that both STD and SFA are capable of inducing bi-directional sweeps. However, (based on our simulations) only SFA can produce uni-directional sweeps. The absence of uni-directional sweeps based on STD may be due to the subtle yet important differences between the two mechanisms. Specifically, STD modulates the neural activity by weakening the recurrent connections, which theoretically can only inhibit recurrent inputs, while SFA can attenuate all forms of excitatory inputs, including external inputs. However, since we did not exhaustively explore the entire parameter space, we cannot conclude that STD is incapable of producing uni-directional sweeps. Future simulations are required.

According to the Reviewer’s suggestion, we added few sentences to discuss the distinctions between STD and SFA in generating theta sweeps in the CANN in line 432 to 440 in the Discussion session:

“Based on our simulation, both STD and SFA show the ability to produce bi-directional sweeps within a CANN model, with the SFA uniquely enabling uni-directional sweeps in the absence of external theta inputs. This difference might be due to the lack of exhaustively exploration of the entire parameter space. However, it might also attribute to the subtle yet important theoretical distinctions between STD and SFA. Specifically, STD attenuates the neural activity through a reduction in recurrent connection strength, whereas SFA provides inhibitory input directly to the neurons, potentially impacting all excitatory inputs. These differences might explain the diverse dynamical behaviors observed in our simulations. Future experiments could clarify these distinctions by monitoring changes in synaptic strength and inhibitory channel activation during theta sweeps.”

(2) I appreciate the inclusion of the experimental data in Fig 6a (though I don't find the left-most panel very useful). I also understand what the authors are trying to convey with plots in 6c and 6c. However, I don't find the text that was added above very helpful at all. I was hoping for a simpler demonstration of the effect, by plotting a series of sequential sweeps (cell index vs time, with color indicating firing rate, as in Fig 2d) in the case of both the slow speed and fast speed regimes. Here, vertical lines could mark the individual theta cycles and the firing of individual cells, showing the constancy of the former but change of the latter.

Thank you for your constructive feedback. It seems there might be a misunderstanding in our previous explanation, for which we apologize. The phenomenon we want to elucidate is not an increase in the theta frequency as detected in LFPs, but rather the slope of phase precession with respect to the animal's movement speed. Due to phase precession, the oscillations of place cells as the animal traverses the field is higher than the theta frequency. A plot as Fig 2.d will not make this point clearer, since it shows the baseline theta frequency (i.e., theta sweeps as we claimed previously). A straightforward way of thinking this point is as we added previously: “…The faster the animal runs, the faster the extra half cycle can be accomplished. Consequently, the firing frequency will increase more (a steeper slope in Fig. 6a red dots) than the baseline frequency”. We hope this clarification addresses the concerns raised.

(3) This is still confusing to me. I just don't understand how the *phase* of the oscillating activity bump has anything to do with the movement of the animal. I would like to see a plot of the sweeps (again, cell index vs time, with color indicating the firing rate) before and after inactivation for short and long duration inactivation. Perhaps I am not understanding or appreciating how the bump recovers after inactivation and how this is related to the motion of the animal.

Thank you for pointing this out again. The activity bump will naturally pop out at the input location (which moves forward than before) after we remove the inactivation and then starts to sweep again as before the inactivation. Single cell phase precession and populational theta sweeps are actually the two sides of the same coin (if all cells start at roughly the same phase in theta cycles). If the reviewer accept this, then at the new location, the activity bump sweeps again (around the new location), and therefore phase precession starts again at a further phase, since phase codes the position as the animal traverses the place field.

(4) I am glad the authors are spending more time discussing this phenomenon, but I am unsure of their explanation: for a sweep moving at constant speed, neurons all along the path will be equally affected (inhibited), so where does the bias for suppressing the "end" neurons come from?

While it may appear that neurons along the path are equally inhibited as the bump sweeps over them, our model incorporates external inputs with Gaussian profiles. These inputs bias neurons closer to the input location, resulting in fewer activations in neurons further away from the input position.

(5) Here I was hoping that the authors might comment on what they suspect happens when the animal starts (or stops) moving, and how the network shifts from tracking regime to oscillatory regime (or vice versa), as is typically seen in experimental data (see for example, Kay et al., 2020, fig 4b,c). My apologies for not making this point clearer.

Thank you for pointing this out. In our model, we observed that when the animal stops, the network continues to generate theta oscillations near the input location, albeit with reduced amplitude (so the network dynamics looks like in the tracking regime). However, we hypothesize that when the animal pauses its movement for enough time (immobile but awake states), sensory input into the hippocampus also decreases, which is similar to removing external inputs in our model. In this case, the activity bump spontaneously moves away, resembling the phenomenon of replay (see also Romani & Tsodyks 2015).

Regarding the experimental data (Kay et al.), it indeed appears that theta sweeps decoded from neural activity become less pronounced when the mouse moves at slower speeds. This observation could potentially correspond to a decrease in the amplitude of bump oscillations when external inputs associated with movement are halted but not entirely removed in our model. However, in experiments, when the mouse's movement slows down, hippocampal activity no longer oscillates at theta frequency, making it challenging to decode theta sweeps.

We appreciate your clarification on this point and recognize the importance of further investigating how our model can accurately replicate the transition between tracking and oscillatory regimes observed in experimental data.